# Flood Risk Mapping by Remote Sensing Data and Random Forest Technique

## Hadi Farhadi [1] and Mohammad Najafzadeh [2,*]

1   Department of Photogrammetry and Remote Sensing, Faculty of Surveying Engineering, K. N. Toosi University of Technology, Tehran 19697-64499, Iran; hadi.farhadiabeshahmadloo@email.kntu.ac.ir
2   Department of Water Engineering, Faculty of Civil and Surveying Engineering, Graduate University of Advanced Technology, Kerman 7631885356, Iran
*   Correspondence: m.najafzadeh@kgut.ac.ir

**Abstract:** Detecting effective parameters in flood occurrence is one of the most important issues that has drawn more attention in recent years. Remote Sensing (RS) and Geographical Information System (GIS) are two efficient ways to spatially predict Flood Risk Mapping (FRM). In this study, a web-based platform called the Google Earth Engine (GEE) (Google Company, Mountain View, CA, USA) was used to obtain flood risk indices for the Galikesh River basin, Northern Iran. With the aid of Landsat 8 satellite imagery and the Shuttle Radar Topography Mission (SRTM) Digital Elevation Model (DEM), 11 risk indices (Elevation (El), Slope (Sl), Slope Aspect (SA), Land Use (LU), Normalized Difference Vegetation Index (NDVI), Normalized Difference Water Index (NDWI), Topographic Wetness Index (TWI), River Distance (RD), Waterway and River Density (WRD), Soil Texture (ST)), and Maximum One-Day Precipitation (M1DP)) were provided. In the next step, all of these indices were imported into ArcMap 10.8 (Esri, West Redlands, CA, USA) software for index normalization and to better visualize the graphical output. Afterward, an intelligent learning machine (Random Forest (RF)), which is a robust data mining technique, was used to compute the importance degree of each index and to obtain the flood hazard map. According to the results, the indices of WRD, RD, M1DP, and El accounted for about 68.27 percent of the total flood risk. Among these indices, the WRD index containing about 23.8 percent of the total risk has the greatest impact on floods. According to FRM mapping, about 21 and 18 percent of the total areas stood at the higher and highest risk areas, respectively.

**Keywords:** Remote Sensing; Google Earth Engine; Random Forest; Flood Risk Mapping





## 1. Introduction

Floods, which are one of the most common types of natural disasters in the world, have a very high potential for destruction [1]. Over the past few decades, the occurrence of numerous floods has had irreversible impacts on the economy, vital resources, and benefits of human beings around the world [2,3]. Therefore, Flood Risk Mapping (FRM) is one of the most important challenges in the assessment of the potential of risks of floods and to consequently reduce their destructive impacts in flood-prone areas. Due to the high number of effective parameters when considering the occurrence of floods, the application of up-to-date and accurate information is of high importance in the decision-making quality. The purpose of flood risk mapping is to accurately manage floods that are caused by rainfall and dam overflows in order to reduce damage to human life and property. Heavy rainfall along with unfavorable environmental and geographical factors such as the topography and its derivatives, Land Use (LU), streams, and rivers as well as the presence of dams in particular, lead to floods in the area. Flood risk assessment is a qualitative or semi-quantitative technique that considers simultaneous effects of various environmental factors such as land topography, soil texture, precipitation, typical LU, and the hydrological properties of watersheds [4]. To simulate flood events, having accurate field information

from the physical characterizations of kinematic waves (i.e., wave velocity, wave height, and flow velocity) that have occurred in the floodplain is a highly time-consuming process and costly task. For this purpose, various numerical models such as Finite Difference Methods (FDMs) and Finite Element Methods (FEMs) have been widely applied to solve governing equations (one of which is known as the Saint-Venante equation) on the flood events in 1-D, 2-D, and 3-D [5–7]. The accuracy level of both FDMs and FEMs is completely dependent on various factors such as the hydrodynamic conditions of the flood, the availability of boundary conditions for solving the governing equations, the availability of recorded information from gauged basins, and the motion of sediments [8–11]. Furthermore, the suitability of typical numerical schemes (explicit or implicit) and grids size for solving flood equations will affect the performance of FEMs and FDMs, respectively. Overall, there is a wide range of factors affecting the precision degree of flood simulation. In comparison techniques such as prototype observations, physical/experimental models, and mathematical techniques, which work based on semi-theoretical and semi-empirical concepts, are restricted to flood scales. In this way, there is an essential need to apply an efficient tool that is less sensitive to these factors. Nowadays, Remote Sensing (RS) has been introduced as a key solution in which the data collection process is performed faster and cost-effectively without human presence in the area [12–16].In the case of flood monitoring, the integration of these data (hydro-environmental indices related to topography, soil texture, rainfall, water bodies density, vegetation situation, and land use) to model risk using Geographical Information System (GIS) software is not conveniently possible despite the need for high computational time. To solve this drawback, a comprehensive web-based platform called Google Earth Engine (GEE) (Google Company, Redlands, CA, USA) can be used [17–23].

The GEE environment is a Cloud Computing Platform (CCP) and online site that hosts global time-series satellite images. It was designed for storing and processing large amounts of data during analysis and decision-making processes [24,25]. All of the geographic data (raw or processed), maps, and tables are freely available from Earth Engine (EE) and can be downloaded to a user's Google Drive (Google Company, CA, USA) or Google Cloud Storage (Google Company, CA, USA) using the GEE platform. The GEE contains a wide range of RS data sets, such as top of atmosphere (TOA) reflectance, surface reflectance, and meteorological data. Several studies have recently been conducted using the GEE platform due to its wide variety of applications and its data processing time [25–27]. The GEE platform can handle massive processes in a short amount of time. The GEE can be widely utilized in a variety of environmental fields, including agriculture, natural resources, and natural disaster monitoring [26,28–32]. Therefore, using this platform, it is possible to produce and combine different models, which can be performed in less time with high speed.

In the case of GEE applications in flood detection, several attempts have been made in which potential applications of this system were studied [24,29,33]. For instance, Liu et al. [34] introduced a rapid flood prevention and response system in the GEE platform that used optical and radar data. One of the issues in flood risk mapping is the combination of various flood indices (e.g., ecological and hydro-environmental factors) and establishing the relationship among them. To reach this goal, systematic methods and Multi-Criteria Decision Making (MCDM) processes such as the Analytic Hierarchy Process (AHP) [35,36], Set Pair Analysis (SPA) [20], and Artificial Intelligence (AI) models have been used successfully. Seejata et al. [22] assessed flood risk areas using the AHP method. To spatially analyze and conduct flood risk mapping, six physical indices including LU, Elevation (El), Slope (Sl), Rainfall, River Density (RD), and Soil Texture (ST) in the ArcGIS (ESRI, Redlands, CA, USA) environment were used and were finally able to be detected in high-risk areas. Bourenane et al. [19] evaluated the FRM in urban areas where they used hydro geomorphological interpretation and analysis methods. Youssef et al. (2019) evaluated the flood risk model using the AHP method and ArcGIS software in the Egyptian region. In this research, to map flood risk, a combination of influential factors was used in the study area.

Finally, they evaluated the proposed model with an Overall Accuracy (OA) of 83 percent. Ogato et al. (2020) assessed flood risk mapping using GIS and MCDM methods with LU, El, Sl, Rainfall, Drainage Density (DD), and ST indices. Guerriero et al. [37] prepared a flood risk mapping analysis that used several possible models using data derived from the Lidar system and hydrometric models. Eini et al. [38] developed a flood risk mapping analysis of urban areas using Machine Learning (ML) techniques. In their research, flood risk maps were generated using two ML models: MaxEnt and Genetic Algorithm Rule Set Production (GARP). They used the economic, social, and infrastructural factors through the optimization process for the flood risk analysis. The Fuzzy AHP (FAHP) method was used for the general weighting of the indices. To evaluate the generated map, Area Under the Curve (AUC) and Receiver Operating Characteristic (ROC) were used, which were equal to 96.76% and 98.32%, respectively. The results of the FAHP model indicated that the MaxEnt technique gave a more satisfying performance compared to the GARP technique. The SPA method also had a significant dependence on the weights of the selected indices and factors, and additionally, it had high complexity and computational time [20]. To enhance the flood risk assessment performance, Machine Learning (ML) methods such as Support Vector Machines (SVM), Gaussian Process Regression (GPR), Decision Trees (DT), Artificial Neural Networks (ANN), Boosted Regression Trees (BRT), Multivariate Adaptive Regression Splines (MARS), and the Group Method of Data Handling (GMDH) were applied [15–17,19,23,39–41]. Among these ML techniques, Random Forest (RF) is an ensemble learning technique that is generally applied for classification and regression performance. RF has been applied to investigate the physical patterns of various processes such as earthquake-induced damage classification [42], rockburst prediction classification [43], tree species classification [44], gene selection [45], and computer-aided diagnosis [46]. Although this ML technique has demonstrated good performance in various applications, less attention has been paid to flood risk mapping [47]. Generally designed for ML classification, the RF model has received a great deal of popularity in RS applications, where it is employed in remotely sensed imagery classification due to its highest precision level compared to other ML methods. Furthermore, the RF model provides the appropriate speed that is required and well-organized parameterization for the process [48]. The subtle differences between the present study and previous literature are the use of upgraded web-based data to produce flood risk indices in which there is no need for powerful PC components to produce effective indices in flood risk analysis. In the case of flood monitoring, the implementation of ML techniques in the GEE platform by upgraded web-based data does not need complex and heavy calculations compared to other platforms such Python (Guido van Rossum, DE, USA), MATLAB (Mathworks, MA, USA), ENVI (L3 HARRIS, Boulder, Colorado, USA), and ArcMap (Esri, West Redlands, CA, USA) softwares. In addition, the GEE platform does not require downloading images and image processing. This issue can be considered as novelties of this research work. Another positive aspect of the present research is that the 11 risk indices, which are listed as El, Sl, LU, RD, ST, Slope Aspect (SA), Normalized Difference Vegetation Index (NDVI), Normalized Difference Water Index (NDWI), Topographic Wetness Index (TWI), Waterway and River Density (WRD), and Maximum One-Day Precipitation (M1DP), are considered for the flood monitoring. Simultaneous usability of these risk indices has not yet been applied in previous literature. It is possible to implement an RF model on the Google Earth Engine cloud platform; however, this process is conducted using the interactive Python and GEE interaction package (geemap: A Python package for interactive mapping with Google Earth Engine) [49]. The reason why this algorithm is not implemented directly inside Google Earth Engine is that the ease of using this engine when integrating various effective indices in creating floods, difficulty in terms of importing samples, ease of quantitatively evaluating the produced model, ease of determining the importance indices as well as ease of outputting raster data. The process of producing a flood risk map in the present paper has not been conducted completely with the GEE platform, and only the flood risk indicators have been extracted in the mentioned platform. Although risk indices are calculated in the GEE environment,

desktop-based software (such as ArcMap [(Esri, West Redlands, CA, USA)]) is used to reclassify and adjust risk indices since one of the disadvantages of the GEE platform is its poor performance when visualizing shapes compared to desktop software such as ArcMap 10.8 (Esri, West Redlands, CA, USA). Therefore, the Python programming language and ArcMap software have been used to implement the Random Forest algorithm to obtain the flood risk map.

The research organization of this work is listed as follows: (i) the second section introduces the study areas and data sets and the GEE, in which two reliable data sources are received from Landsat 8 (L8) satellite imagery and the Shuttle Radar Topography Mission (SRTM) Digital Evaluation Model (DEM); (ii) detailed information about the proposed methodology is provided in the third section; (iii) the results of methodology implementation for various risk levels are provided. Then, the results of this study are compared with those from the literature; and (iv) the key achievements are presented in the conclusions.

## 2. Overview of Case Study and Data

### 2.1. Research Case Study

The study area used for this research is the Galikesh basin. The Galikesh basin, which is located in Golestan province and is derived from a sub-basin of Gorganrood, has an area of 404.80 square kilometers. The maximum and minimum heights of the basin are 2461.3 m and 378.1 m, respectively. The average height of the basin from sea level is 1395.2 m, and the average slope of the basin is also equal to 23.3 percent. An overview of the case study is illustrated in Figure 1. Due to the location of this river and the corresponding expected flooding in recent years, parts of the North Khorasan and Semnan provinces were also affected by the flood. This case study includes an area of 620 square kilometers with dense vegetation, which is located in moderate topographic conditions and is at an altitude of about 1500 m above sea level. Galikesh city has sufficient rainfall quantity due to its geographical location. In this way, there are surface water networks and storage of groundwater resources for local dwellers in the cold and warm months of the year, respectively. Lack of access to the surface water network in some parts of this region has had its inhabitants benefit from groundwater through the construction of aqueducts and various deep and semi-deep wells. In this region, one of the natural disasters that has put surface water and groundwater resources in a dangerous state is flood events. Floods that occur in the area could have detrimental impacts on the quality of water resources, agricultural and human activities, and livestock losses. Four flood events that occurred in the past (12 August 2002; 13 September 2008; 15 October 2014; and 12 July 2020) had severe impacts on various environmental aspects of the area, such as deforestation, the overuse of the region's soil, the abrupt erosion of soil, landslides, sharp elevation directions along the river, and the pollution of water resources. In this way, the assessment of flood risk is essential to ameliorate these repercussions. Given the occurrence of numerous floods and the threat to natural resources, efforts to prevent the occurrence of destructive floods in this basin are inevitable.

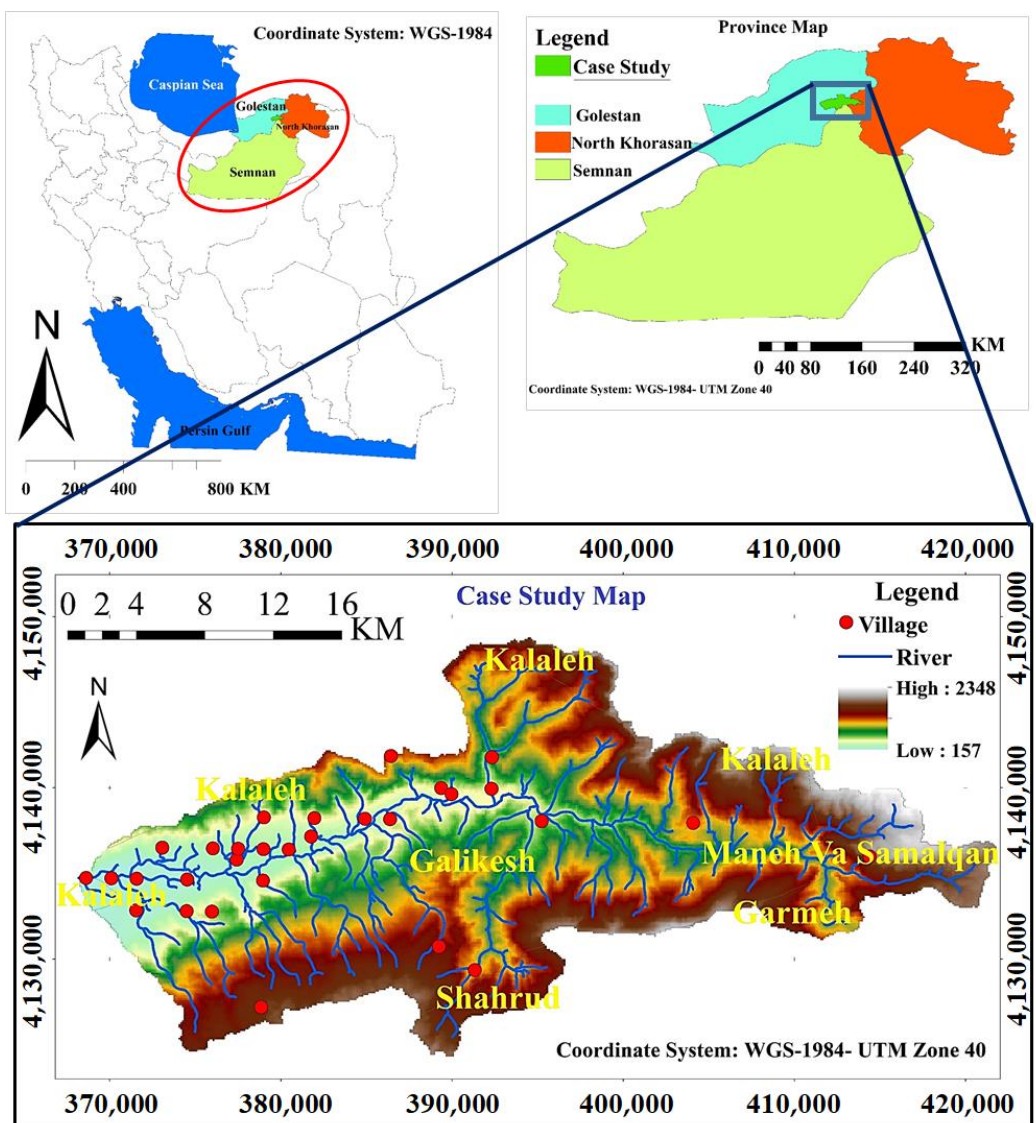

**Figure 1.** Overview of case study area.

*2.2. Data*

In the current study, L8 satellite images and SRTM DEM were used in the GEE cloud platform. In the GEE platform, all of the preprocessing, such as Radiometric and Atmospheric corrections, which have already been performed, reduce the essential time needed to prepare the data for the main analysis.

2.2.1. SRTM DEM

The SRTM DEM is an international project spearheaded by the United States National Aeronautics and Space Administration (NASA) and the United States National Geospatial-Intelligence Agency (NGA). SRTM includes a chiefly modified radar sensor that flew onboard the Space Shuttle Endeavour during the 11-day STS-99 mission in February 2000. The SRTM DEM has a spatial resolution of 30 m and a height accuracy of 16 m [50,51]. The current data obtained from SRTM DEM dates back to a 10-day period (from 11 February 2000 to 22 February 2000), and additionally is freely available in GEE and on the United States Geological Survey (USGS) website (see https://earthexplorer.usgs.gov/).

2.2.2. Landsat 8 Satellite

The L8 satellite is one of the optical satellites launched in February 2013 and has been frequently used in environmental studies. In the present study, the L8 satellite with OLI sensor was used. This sensor consists of nine spectral bands in the panchromatic, visible, and near-infrared bands with a spatial resolution of 15 to 30 m and two thermal bands with a spatial resolution of 100 m and a temporal resolution of 16 days. In the present study, bands with a spatial resolution of 30 m have been used. This satellite data covers the entire globe and can be analyzed for free by GEE and can also be downloaded from the USGS website. Cloud coverage is one potential issue when using L8 Optical data. Therefore, images with less cloud cover are used in the present experiment. Since risk indices are used in preparing flood risk maps, the accuracy and quality of the data used in the production of indices will affect the accuracy of the generated map. In fact, the higher the spatial resolution of the data, the more accurate the produced map. In the present study, images that had less cloud cover over time (below 5%) were used. The characteristics of L8 satellite bands are presented in Table 1. In the present study, the mentioned data (L8 and SRTM DEM) used to generate risk indices were resampled to a 30 m spatial resolution to ensure that all of the products had the same pixel size. Then, the flood risk indices were reclassified.

Given that the management of qualitative and nonlinear data is one of the most significant challenges in the FRM, the first step in flood risk mapping is to select the most important indices affecting flood-prone areas. On the other hand, all flood-prone areas do not have exactly the same characteristics; therefore, these indices may vary in different areas. Therefore, in the present research, according to the location and conditions in the study area, 11 environmental indices (El, Sl, SA, LU, NDWI, NDVI, TWI, RD, WRD, ST, and M1DP) affecting the flood occurrence were used. The flood risk mapping process is presented in Figure 2. According to Figure 2, in order to prepare a flood risk mapping, various data such as L8 satellite imagery and the SRTM DEM model are called upon and are pre-processed on the GEE platform. Then, Landsat 8 satellite images are used to generate four indices: NDVI, NDWI, ST, and LU. Additionally, the SRTM DEM model is used to produce six indices Sl, SA, El, RD, WRD, and TWI. In addition, one-day precipitation data related to CHIRPS satellite are used. Therefore, in general, 11 risk indices are generated and are used to model flood risk. In the next step, the RF model is fed by the values of each risk index, which are obtained from historical floods, for the performance of the training stage. After evaluating the RF model by the testing data, a flood risk map is generated for the values of all of the pixels and then the importance of each index is determined.

**Table 1.** Details of Landsat 8 satellite bands.

| Satellite/Sensor | | Band Name | Wavelength | Resolution |
|---|---|---|---|---|
| | Band-1 | Coastal/Aerosol | 0.43–0.45 | 30 |
| | Band-2 | Blue | 0.45–0.51 | 30 |
| | Band-3 | Green | 0.53–0.59 | 30 |
| | Band-4 | Red | 0.64–0.67 | 30 |
| | Band-5 | Near Infrared (NIR) | 0.85–0.88 | 30 |
| Landsat 8/OLI | Band-6 | Shortwave Infrared (SWIR) 1 | 1.57–1.65 | 30 |
| | Band-7 | Shortwave Infrared (SWIR) 2 | 2.11–2.29 | 30 |
| | Band-8 | Panchromatic | 0.50–0.68 | 30 |
| | Band-9 | Cirrus | 1.36–1.38 | 30 |
| | Band-10 | Thermal Infrared (TIRS) 1 | 10.6–11.19 | $100 \times (30)$ |
| | Band-11 | Thermal Infrared (TIRS) 2 | 11.5–12.51 | $100 \times (30)$ |

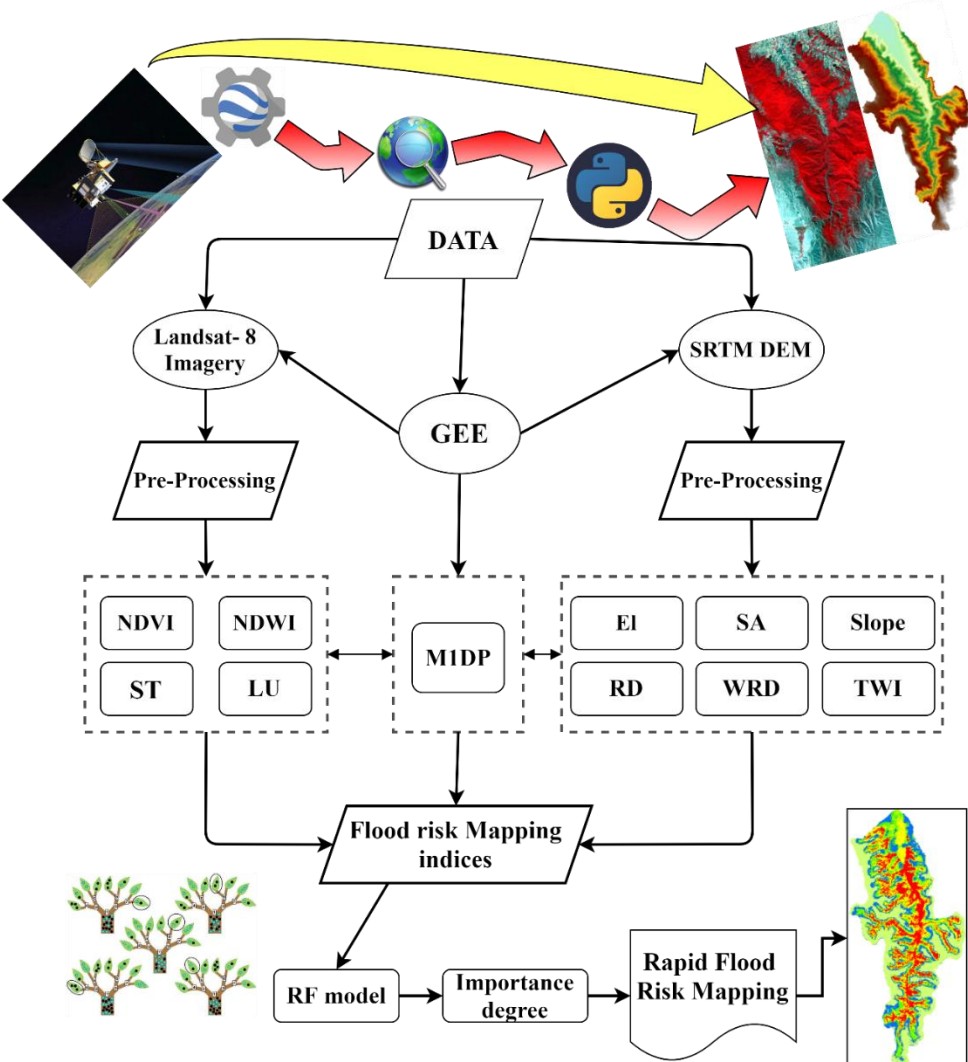

**Figure 2.** Schematic diagram of rapid flood risk mapping.

## 3. Methodology of the Research

### 3.1. Definition of Indices

According to Figure 2, the SRTM DEM and L8 satellite images in the GEE platform were applied to map flood risk. As previously mentioned, in this study, 11 environmental indices influencing the occurrence of floods were used, in which 4 indices produced from L8 satellite images and 6 indices produced from SRTM DEM were used.

All of the indices that were extracted from L8 satellite imagery and SRTM DEM in the GEE web-based platform were imported into the ArcMap 10.8 (Esri, West Redlands, CA, USA) software to analyze and prepare the graphical output. In the next stage, the appropriate values of the training and testing samples were extracted from these indices, which were then imported into the Python programming environment. Finally, after predicting various risk levels, the predicted values were imported into the ArcMap 10.8 (Esri, West Redlands, CA, USA) software, and the final flood risk model was created.

### 3.1.1. Elevation

The SRTM DEM model presents the point elevation changes in the study area in each pixel, which is in the unit of meters. The SRTM DEM model was provided by the GEE platform, and the elevation point values are between 157 and 2348 m for the case study. In the case of flood risk, points that have higher elevations are less susceptible to risk than other points with low elevations. For instance, the point placed on the summit of the mountain has a lower risk than the points on the hillside.

### 3.1.2. Slope

The amount of slope in each pixel relative to the surface level obtained from the DEM and the value of the pixels vary between 0–100 percent. Due to the mechanisms of water and its flow in areas with sharp slopes and its gentleness in flat areas, this index plays a significant role in monitoring floods in degree units. For the present case study, the slope values were 0–69.98%. The higher the slope values, the higher the flood risk.

### 3.1.3. Slope Aspect

This index is also a derivative of the DEM that determines the directions of the pixels per unit degree, leading to more accurate decisions in flood risk mapping. In other words, the SA map is the slope deviation from the geographical north, which varies between 0 and 360 degrees. In fact, determining the SA in the direction of a slope and with a certain height is capable of completely modeling the shape of the earth. For the SA index, there are four main directions and four inter-cardinal directions. Hence, there are eight classes for SA features: N(1), NE(2), E(3), SE(4), S(5), SW(6), W(7), and NW(8). In flood risk mapping, the number of identification classes is directly assigned to each satellite imagery pixel.

### 3.1.4. Land Use

The occurrence of floods in areas with different characteristics shows different performance. For instance, the flooding mechanism is very different in urban areas, areas with vegetation, and areas with soil and rock texture. Therefore, each feature will play a substantial role in flood risk mapping. In the present study, weight different types of LU features (i.e., water, urban, bare area, cropland, agriculture land, shrub, forest, and herbaceous area) were defined in the case study, including 70 percent of the medium-density and dense vegetation and 30 percent of the vegetation in other areas. Runoff coefficients related to each LU feature are presented in Table 2. As seen in Table 2, water and shrub have the maximum and minimum runoff coefficients, respectively. An increase in the values of the runoff coefficient increases the flood risk. Then, eight various classes were assigned for LU, features as seen in Table 2. In this study, the various runoff coefficients were considered based on two Chinese standards: (i) the code for design for the construction of water supply and drainage (GB 50015-2003) and (ii) the code for the design of outdoor wastewater engineering (GB50014-2006). The LU index was generated using the time-series images from L8 satellite in the GEE, in which SVM, as a supervised classification model, was used to classify LU with an OA of 95 percent.

**Table 2.** Land use pattern and corresponding runoff coefficients in the Galikesh River Basin.

| Land Use | Forest | Shrub | Herbaceous | Agriculture Land | Cropland | Bare Area | Urban | Water (River) |
|---|---|---|---|---|---|---|---|---|
| Runoff Coefficient | 0.15 | 0.18 | 0.2 | 0.4 | 0.6 | 0.7 | 0.9 | 1 |
| Identification Number of Class | 1 | 2 | 3 | 4 | 5 | 6 | 7 | 8 |

### 3.1.5. Normalized Differential Vegetation Index

Although the location of vegetation is clear in the land use map, the vegetation status in terms of density and canopy is not clear. To accurately calculate the amount of vegetation (canopy), the Normalized Difference Vegetation Index (*NDVI*) was used and could be calculated as

$$NDVI = \frac{NIR - R}{NIR + R} \tag{1}$$

In Equation (1), the *NDVI* is the Normalized Differential Vegetation Index, *NIR* is the amount of near-infrared band reflectance, and *R* is the red band reflectance of the L8 satellite imagery. Using this index, the vegetation values can be extracted in different intervals. The value of the *NDVI* varies between +1 and −1, in which values ranging from 0.2 to +1 are classified as vegetation class [52]. In this research, five classes were defined for the vegetation situation in the case study. Bare area, water, low vegetation, medium vegetation, and high vegetation were assigned to the *NDVI* map by classes 5 to 1, respectively. Clearly, flood risk in the bare area is higher than it is in districts where there are various vegetation densities.

### 3.1.6. Normalized Difference Water Index

One of the significant elements in flood risk mapping that has not drawn great attention is the presence of water bodies such as water reservoirs, dams, rivers, and permanent water [17,46,53]. These areas are always a threat to neighboring lands. In the case of a threat, these water bodies will cause the soil to become as saturated as possible, and the potential for a flooding event occurring is unavoidable. Since the *NDWI* is a dynamic index that can be used over time and since risk is a static phenomenon, the average amount of water areas (Mean (*NDWI*)) in 12 months was used. Since Landsat 8 satellite images are taken twice a month and since the water zones change at different times, the average monthly value of the water zone was used to have the best estimate of the amount of water in the study area. The *NDWI* index has been used to separate water class from other classes, such as soil, vegetation, etc. The index was introduced by McFeeters [54] to determine water characterization using the Landsat-TM Green and *NIR* Band (Band 2 and Band 4). Positive values in the *NDWI* image were classified as water classes, whereas negative values were identified as non-water areas [54]. This index can be obtained as follows:

$$NDWI_j^i = \frac{G - NIR}{G + NIR}, NDWI_{Ave} = Mean(NDWI)_j^i \tag{2}$$

where $NDWI_j^i$ is the monthly Normalized Differential Water Index, $i, j$ is the first to the last monthly NDWI, $NDWI_{Ave}$ is the average of the monthly *NDWI*, *NIR* is the amount of near-infrared band reflectance, and *G* is the green band reflectance of the L8 satellite imagery. Using this index, the water values can be extracted at different intervals. The *NDWI* values between zero and one are considered as water areas. Areas that are covered by water have a higher flood risk compared to the districts that are not covered by water. In this study, the distribution of the *NDWI* values provided two classes (0 for non-water area and 1 for water).

### 3.1.7. Topographic Wetness Index

This index can be generally applied to measure the number of topographic controls on hydrological events. TWI is in close connection with the slope and the upstream, contributing an area per unit width that is orthogonal to the direction of flow. As mentioned in [55], TWI is generated based on its relationship definition and the ramifications of DEM, slope, the direction of flow, and the accumulation of flow when using the raster calculator tools in the ArcMap 10.8 (Esri, West Redlands, CA, USA) software. In the present study, TWI values vary between −5 and 7, indicating low- and high-risk levels, respectively.

### 3.1.8. River Distance

This index is introduced as one of the most significant parameters that can be applied in flood risk mapping. The river system is obtained on the basis of SRTM DEM. The rivers are therefore set to zero; thereafter, this value becomes greater as the distance to the river increases. The RD feature ranges from 0 to 2302.17 m in our case study of the Galikesh River.

### 3.1.9. Waterway and River Distance

WRD, as an important index in flood risk mapping, is quantified by various geological factors: the permeability of areas, the vegetation state of areas, the slope of the surface, and the time duration of the flood/inundation. Generally, there is an inverse correlation between the WRD index and the permeability of different areas. Higher values of WRD denote a high volume of run-off per basin along with the erodible soil texture of different areas (e.g., alluvial sediments, sand, clay), leading to less flood-prone areas [21]. Therefore, the rating for the WRD index has an inverse correlation with the WRD index. The WRD index can be computed as

$$WRD = \frac{L}{A} \tag{3}$$

where $L$ and $A$ are the total distance of the water and river channel (km) and the total area of the watershed ($km^2$), respectively. When $WRD$ is equal to 0, the areas with the lowest risk state are met, whereas for $WRD > 0$, high risk is acquired. The maximum $WRD$ value obtained for the Galikesh River is 2.53, indicating the highest level of risk.

### 3.1.10. Soil Texture

This index, which has an important effect on the occurrence of floods, indicates the type of soil particle characteristics that are present in the case study. As seen in Table 3, according to the codes of each texture, when the classification values of the different classes have larger values, it indicates a high degree of infiltration. These values were determined based on data from the Harmonized World Soil Database [56] and the USDA Soil Taxonomy (ST). From Table 3, it can be inferred that sandy clay with higher infiltration has the highest level of flood risk compared to silt (6) and clay (3) soils. The identification number of the soil classes is dedicated to each pixel used for the flood risk assessment by the RF model.

**Table 3.** Soil texture and corresponding class of infiltration capability.

| Type | Alfisols | Entisols | Mollisols |
|---|---|---|---|
| Texture | Silt | Clay | Sandy clay |
| Indentification Number of Class | 6 | 3 | 8 |
| Infiltration Level | Moderate | Low | High |

### 3.1.11. Maximum One-Day Precipitation

Using different satellites by means of the Cloud Computing Platform, the GEE platform can extract the precipitation trend changes that are available in 30-min, 3-h, daily, and monthly intervals. Climate Hazards Group InfraRed Precipitation with Station data (CHIRPS) are 30+ year quasi-global rainfall data. CHIRPS incorporates 0.05-degree spatial resolution satellite imagery with in situ measurement data to create gridded rainfall time series for trend analysis and seasonal drought monitoring. In the current research, four flood events take place on four separate days; then the Maximum One-Day Precipitation (M1DP) index was used to analyze flood monitoring. The consideration of the maximum precipitation is the more persuasive rainfall index than other precipitation values for providing flood risk mapping. Then, the M1DP values of the stations are interpolated using the Kriging interpolation tool, and the network layer of the M1DP index can finally be generated (30 × 30 m) based on the ArcGIS (ESRI, Redlands, CA, USA) software. From

meteorological information, M1DP values vary between 37.29–73.84 mm for the four previous flood events seen in the study area. The lower M1DP values generate a higher level of flood risk.

### 3.2. Collection of Training Data

The collection of the sample data is of high importance when mapping the flood risk to evaluate the performance of the testing and training sets, which has a significant impact on the quality and reliability of the output map. In the current study, according to the recent floods that have occurred in the study area, 400 sample data were collected from the high-resolution satellite imagery source and field observation data to develop the RF model. These samples were taken from flooded areas and non-flooded areas. There are 11 risk indices, which are considered input variables, that were used to feed the RF model. The output variable has two labels: non-flooded areas (0) and flooded areas (1). In fact, 400 sample data were equally selected from two typical areas. From all of the sample data, 70 and 30 percent of the sample data were considered for the training and testing stages, respectively. In this study, the training and testing sample data were normalized between 0 and 1 prior to being fed by the RF model.

In this study, data were associated with the floods that occurred on 12 August 2002 (lower risk), 13 September 2008 (medium risk), 15 October 2014 (higher risk), and 12 July 2020 (highest risk). All of the sample data at the risk state were grouped into the flooded area. According to the data, the samples in which floods did not occur are considered to be the areas with the lowest risk level. Moreover, this typical selection of risk categories was used by previous investigations in which the flooded areas at the flooding time were classified as the areas that are at medium and high risk [17,57,58]. The number and distribution of the sample data in the study area are shown in Figure 3. In Figure 3, the training sample data are uniformly distributed throughout the study area.

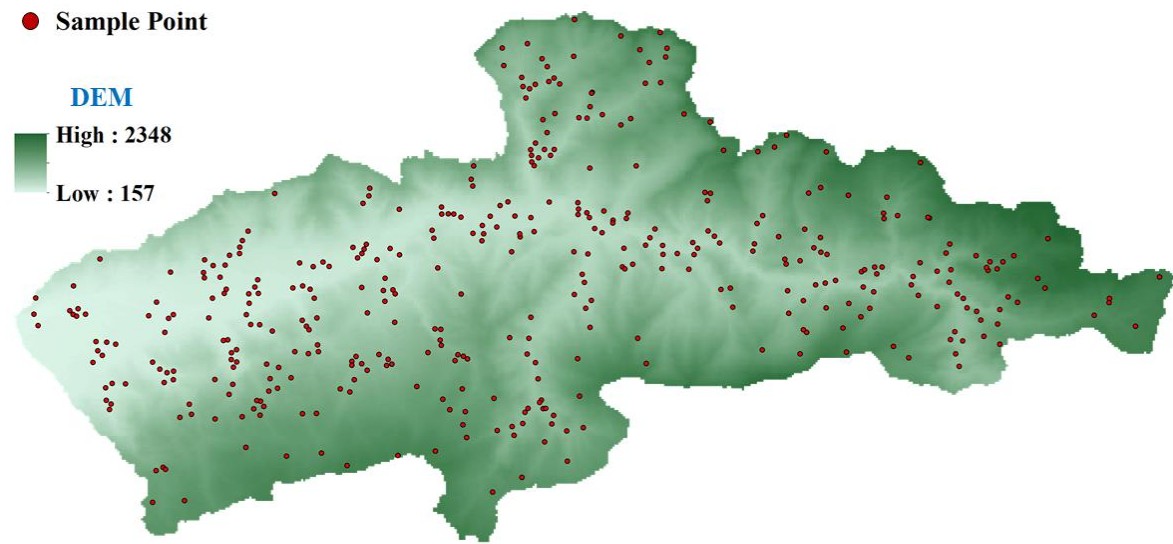

**Figure 3.** Distribution of data samples in the case study.

### 3.3. Implementation of Random Forest

The RF model, as one of the most well-known supervised classification methods, is capable of combining the predictive trees and was proposed by [59]. In the RF method, each tree is dependent on the values of a random input vector independently and has equal distribution to all of the trees in that forest [59]. Currently, this algorithm is a widely used classification algorithm for high-dimensional data as well as multi-modal data classification among classification algorithms [57,60]. The RF algorithm is highly useful to reduce the frequently reported overfitting cases that are associated with the Decision Tree (DT) model. This non-parametric algorithm is based on DT algorithms. DT is a hierarchical classification

algorithm that can be used to label an unknown pattern by using a sequence of decisions. The trees include three main elements: root nodes, child nodes, and leaf nodes (terminal nodes). The implementation of classification is performed by using a set of rules that define the path to be followed, starting from the root node and ending at one terminal node, which indicates the label for the classification of the object. At each non-terminal node, a decision is made regarding the path to the next node [61]. The flowchart of the RF model is conceptually sketched in Figure 4.

In the first step, the $k$ subsets of the training data ($D_1$, $D_2$, ..., $D_k$) are selected from the entire training data ($D$) by means of the Bootstrap Sampling (BS) technique. The BS is a statistical technique for estimating the quantities of a population by averaging estimates from multiple small data samples. Importantly, samples are created by drawing observations from a huge data sample one at a time and returning them to the data sample after they have been selected. Additionally, the sample size of $D_k$ is the same as the whole of sample set $D$. Next, $k$ DTs are generated based on the $k$ subsets and $k$ classification results. At the final step, each DT casts a unit vote for the most popular class; therefore, the most satisfying results are acquired.

The typical risk indices (11 risk indices) are not selected randomly while the values of the risk indices are selected based on the principles of the RF model. The accuracy of the generated flood risk map was evaluated using the collected samples, which are considered as samples for the training (70 percent) and testing (30 percent) data. In the case of RF performance, there are setting parameters that may produce uncertainty in the prediction accuracy level. These parameters are the number of estimators, the maximum number of features considered for splitting a node, the maximum number of levels in each DT, and the minimum number of data points put in a node prior to the splitting node. In some performances, the general structure of the RF model is overparameterized, requiring an overfitting reduction (when the training stage is too much accurate and the testing result is too inaccurate) by using the $k$-fold cross-validation technique. The overfitting of the RF results produces huge uncertainty. In this study, to avoid the possibility of overfitting, we assigned five-folds during RF performance among other $k$-fold values. Similar investigations that have used ML approaches have proven that five folds are generally a sufficient number of folds to reduce overfitting occurrence [17,43,44,57]. After performing the model in the training and testing phases, the RF model was developed for all pixels (206216). In fact, each pixel includes 11 features (risk indices) and 1 label (0 or 1). Once the RF technique was performed for all of the pixels, the risk values were obtained between 0 and 1. These values were classified into five classes: lowest risk, lower risk, medium risk, higher risk, and highest risk.

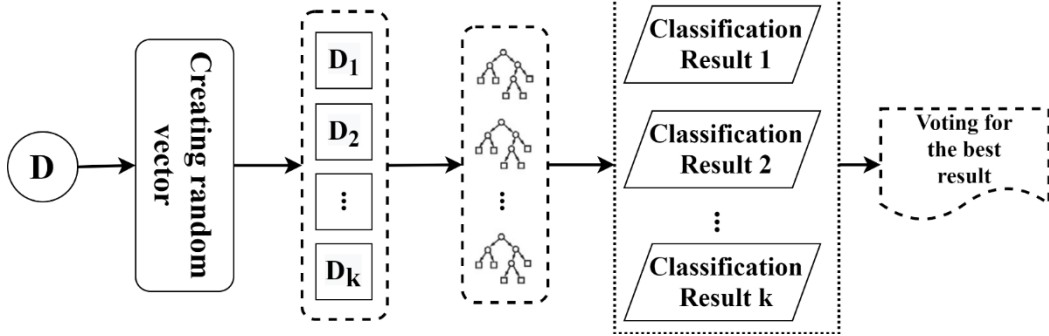

**Figure 4.** Conceptual scheme of RF model for classification process.

### 3.4. Flood Risk Mapping and Risk Assessment

To provide the map of the flood, the risk indices generated in the GEE platform (11 indices) were converted to raster grids, and each risk index had a 206216 raster grid. Then, according to the floods that had occurred in previous years in the study area, a set of training data was prepared. As mentioned in the previous section, these sample data were partitioned into two main sections to create the RF model and to assess the accuracy level of the output map. In the next step, the Importance Degree Index was determined while creating the model in the Python package (i.e., seaborn, numpy, pandas, and matplotlib). After creating the final map, the trained model was evaluated using the test data in the study area.

### 3.5. Definition of Error and Index Importance Degree

For a specific training set ($Q$) with a sample size $M$, the quantity of probability that each sample data in the set $Q$ would not be acquired is $[1 - 1/M]^M$. Once the sample size $M$ is adequately large enough, the probability value increases to 0.368. This value indicates that more than $1/3$ of the $M$ samples in set $Q$ are placed out of the bootstrap sample; these samples are then called out-of-bag (OOB) data. Generally, OOB data are applied to create a running, unbiased estimate of the classification error as trees are merged into the forest.

The RF algorithm calculates the importance of the variables, which is called the Index Importance Degree (IID). This parameter permits the decision-maker to understand and predict an index contribution to the total risk. Overall, two techniques are used to calculate IID. At first, the OOB error associated with each tree ($E_{OOB1}$) is initially computed; thereafter, these initial values merge the noise to the data of the index $i$, and then the OOB error ($E_{OOB2}$) values are computed. The IID $i$ is acquired by considering the average of the difference between the $E_{OOB1}$ and $E_{OOB2}$ values; thereafter, the standard deviation normalizes this index. In the case of the second technique, a node split is generated on the index $i$ each time, and the benchmark of Gini impurity associated with the two descendent nodes is lower than that of the parent node. The combination of the Gini decreases for each index over all of the trees in the forest, which expeditiously creates an index that is generally consistent with the permutation importance measure [62]. In the current research, the $E_{OOB2}$ technique is applied to calculate the importance degree of every index; afterward, the contribution of each index to the total risk is acquired as

$$p_k = \frac{\sum\limits_{i=1}^{n} \sum\limits_{j=1}^{t} D_{Gkij}}{\sum\limits_{k=1}^{m} \sum\limits_{i=1}^{n} D_{Gkij}} \tag{4}$$

where $m$ is the number of indices, $n$ is the number of classification trees, and $t$ is the number of nodes in the tree structure. Additionally, $D_{Gkij}$ is the value of the decrease in the Gini Index that is associated with the $i$th node in the $j$th tree, which is associated with the $k$th index, and $P_k$ denotes the contribution degree of $k$th index out of all of the existing indices.

### 3.6. Evaluation of the Proposed Model

In order to assess the accuracy and quality of the model produced with real data, Area Under the Receiver-Operator Characteristic Curve (ROC-AUC), Kappa Coefficient (KC), and OA were used. Overall, the rate of predictions indicates the potential of the predictive models in a specific area using these criteria [54,63,64]. Additionally, in order to evaluate the performance of the model results, two well-known statistical benchmarks, Root Mean Square Error (RMSE) and Mean Absolute Error (MAE), were used [65–67],

$$RMSE = \sqrt{\frac{\sum\limits_{i=1}^{N} (Flood_{Pre}^i - Flood_{Obs}^i)^2}{N}} \tag{5}$$

$$MAE \ = \ \frac{1}{N} \sum_{i=1}^{N} \left| Flood^i_{Obs} - Flood^i_{Pre} \right| \tag{6}$$

where $Flood_{Obs}$ is the values of the observed flood data, $Flood_{Pre}$ is the estimated values, and $N$ is the total number of measured data.

In the following, brief descriptions of ROC-AUC are given:

An ROC graph is introduced as a curve illustrating the performance of a classifier technique in all classes. Two parameters, known as True Positive Rate (TPR) and False Positive Rate (FPR), require the ROC curve to be drawn. Additionally, AUC is a statistical measure to assess the capability of a classifier model to distinguish among classes and can be applied as a summary of the ROC graph. These parameters are computed as

$$TPR \ = \ \frac{TP}{TP + FN} \tag{7}$$

$$FPR \ = \ \frac{FP}{TP + FN} \tag{8}$$

The increase in AUC values indicates the more efficient performance of the classifier model in distinguishing between the positive and negative classes. When the AUC is equal to 1, the classifier model is capable of perfectly distinguishing between all of the positive and the negative class points correctly. When the AUC value becomes zero, the classifier model will predict all of the negatives as positives and all positives as negatives. For AUC = 0.5–1, there is a high possibility that the classifier model is capable of distinguishing the positive class values from the negative class values. In the case of AUC = 0.5, the classifier model is unable to make a distinction between positive and negative class points. Other descriptions of ROC–AUC can be found in the literature [38].

## 4. Result and Discussions

### 4.1. Results of Risk Indices

One of the most important advantages of GEE compared to other software is its ability to generate various parameters in a short time and at a high speed. As mentioned, according to the location and conditions in the study area, 11 environmental parameters indices affecting the occurrence of floods, El, Sl, SA, LU, NDWI, NDVI, TWI, RD, WRD, ST, and M1DP indices, were generated from L8 satellite imagery and the SRTM DEM, as shown in Figure 5. As more clarifications for risk indices arise, the link of the script that was used to specifically create the 11 risk indices was provided in the GEE platform. This script can be found in the Appendix A section.

According to Figure 5, all of the effective indices for the creation of floods show the values and density of classes in the study area (especially vegetation class and water areas) with very high accuracy. However, in previous studies, many classes (especially vegetation and water) have been ignored. This is because the density of the vegetation is very effective in creating floods. In other words, the denser the vegetation in the area is, the lower the probability of flooding.

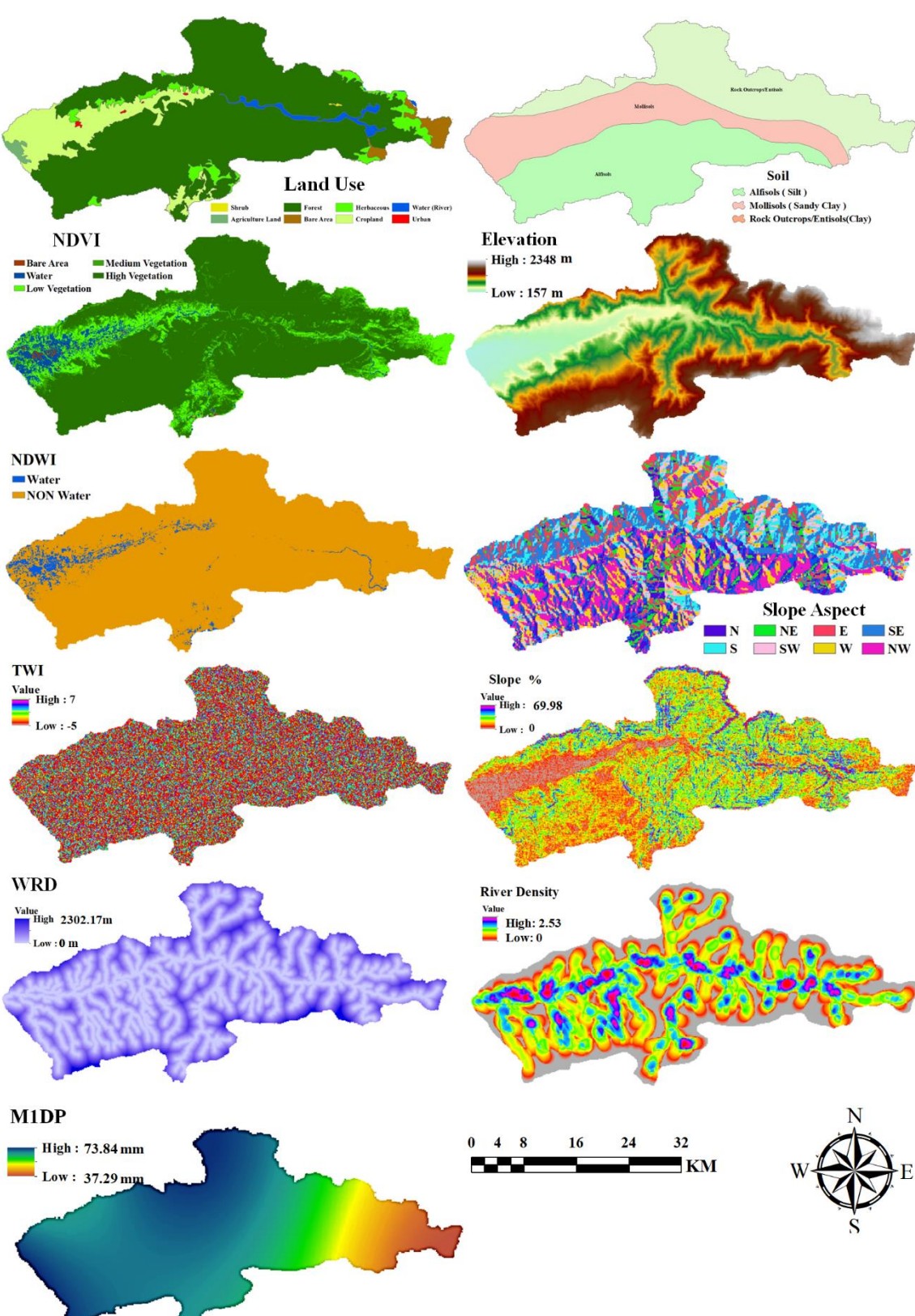

**Figure 5.** Characteristic distributions of risk indices.

*4.2. Flood Risk Mapping*

With the aid of the natural break classification method in ArcMap 10.8 (Esri, West Redlands, CA, USA) software, the five risk classes were assigned to map the flood probability: lowest (0.0–0.2), lower (0.2–0.4), medium (0.4–0.6), higher (0.6–0.8), and highest (0.8–1.0) risks. By combining different layers of indices produced in the GEE and classifying them based on the RF algorithm, the final flood risk mapping was produced in Figure 6, and finally, the highest and lowest risk areas were determined. As shown in Figure 6, the most dangerous area is located at the center of the waterway network in low elevation regions and in the east–west direction. Additionally, in Figure 6, several residential areas are in high-flood risk areas. According to our results, the margins of the waterways and rivers that have higher elevation are the lowest, lower, and medium risk, indicating the importance of EI, SL, and SA. According to the flood risk mapping and risk class distribution, areas with high WRD and RD that are close to the river and that have flat elevation with low vegetation have the highest risk of floods and inundation. In the eastern region of the study, the indices for low elevation, moderate slope and slope aspect, low vegetation, water area, and soil texture are very influential indices when considering the occurrence of floods. In contrast, the lowest risk areas are the areas with dense vegetation, low rainfall values, high elevation, gentle slope, soil texture with high infiltration, and arid areas. Moreover, high values in the rainfall index indicate an increase in flood risk, which has a significant effect on the occurrence of floods. According to Figure 6, about 18% of the study area has the highest risk, 21% is at higher risk, 16% is at medium risk, 16% is at lower risk, and 29% has the very lowest risk. In the study area, there are residential areas that are located in areas with the lowest and highest risk. Due to this issue, with this method, suitable and unsuitable areas for building construction can be located in less time. According to the FRM, about 50 percent of residential areas are located in high-risk areas that would lead to severe losses of life and property in the event of a flood. The location of several examples of recent flood damage in the study area is shown in Figure 7.

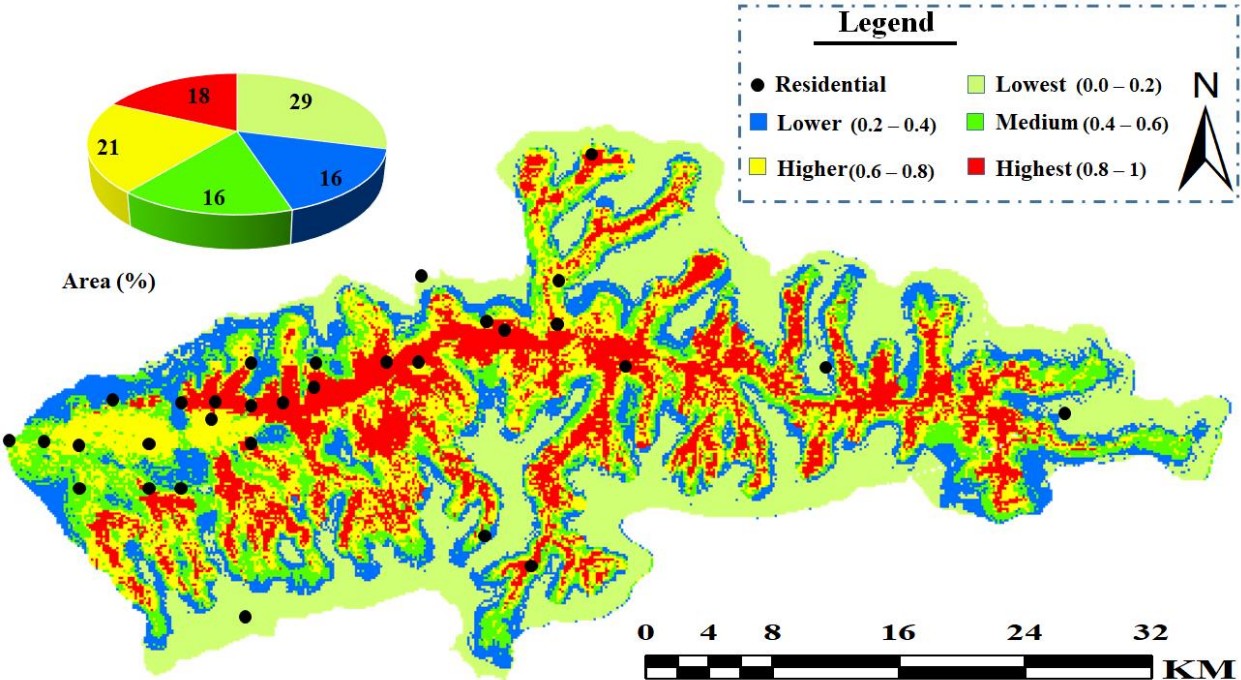

**Figure 6.** Map of flood risk assessment by the RF model.

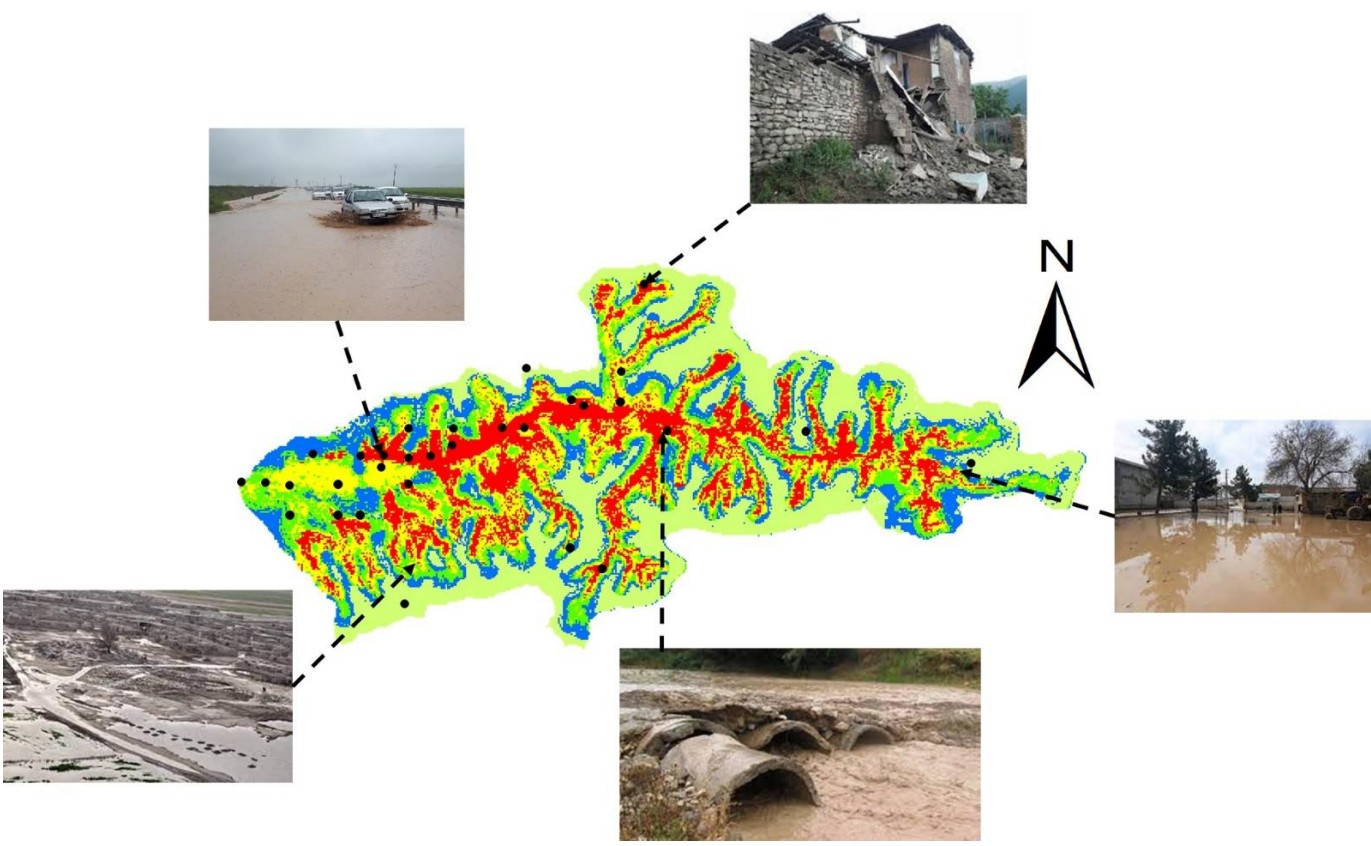

**Figure 7.** The location of several examples of recent flood damage in the study area.

### 4.3. Index Importance Degree Analysis

As mentioned in Section 3.3, the RF computes the variable IID that helps the decision-maker to realize and estimate an index's contribution to total risk. Generally, there are two approaches to compute the IID. The first approach initially computes the out-of-bag (OOB) error of each tree (EOOB1) and then adds the noise to the data of index *i* and subsequently calculates the OOB error (EOOB2). The IID *i* is obtained by taking the average of the difference between EOOB1 and EOOB2 variables and then normalizing it using the standard deviation. In the second approach, when a node split is made on index *i* at each time, the Gini impurity criterion for the two descendent nodes is less than that of the parent node. Combining the decreases in the Gini Index for each individual index over all trees in the forest rapidly provides an important index that is typically consistent with the permutation importance measure. This study adopts the latter approach to compute the importance degree of each index. As such the RF model describes variable importance by enabling an assessment of the importance of each variable using the Gini decrease index. One of the RF algorithm capabilities is to provide the impact of each risk indices using the Gini Index. Based on this index, the importance of each index is estimated in percent, and users and planners use them to provide the appropriate program to reduce risk damages. This capability, provided by Breiman [59], is available in most open-source softwares. Therefore, the impact values of each index were calculated in Python open-source Python (Guido van Rossum, DE, USA) software, which is shown in Figure 8.

According to Figure 8, the indices of WRD, RD, M1DP, and El, account for about 68.27% of the total risk of flooding. This suggests these special indices contribute overwhelmingly to total flood risk. Among these indices, the WRD index, with about 23.8% of the total risk, had the greatest impact on floods. The second risk index is the RD index with a risk of 15.3%. The lowest percentage of flood risk is related to the NDWI index, which accounts for only 1.03% of the total risk. However, NDWI, ST, TWI, SA, LU, NDVI, and Sl indices account for about 31.73% of the total risk. Together, these indices lead to floods and

irreparable damages to agricultural land and crops as well as economic resources. Due to the homogeneity and heterogeneity of features in satellite imagery and DEM, each of the risk indices may present various results in different regions. Therefore, the results of this section show the capability of the RF algorithm to identify the most important and least effective risk factors. The RF model has a significant impact on data generation time and final map modeling. It also increases the quality of decision-making to prevent a disaster.

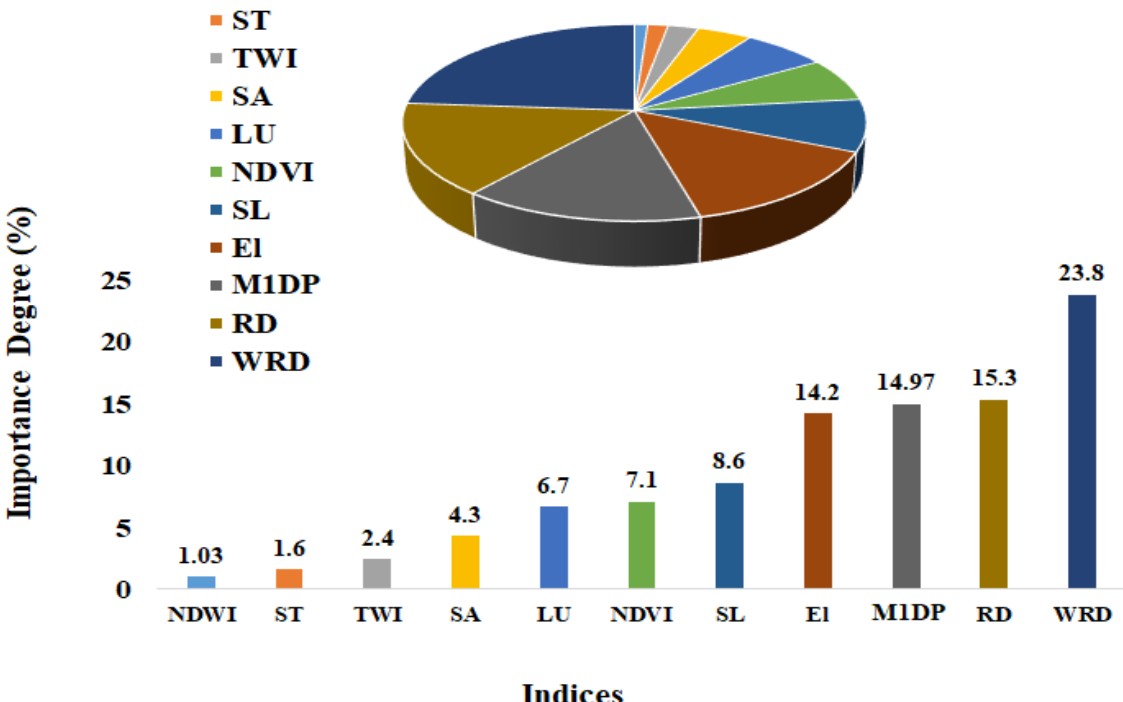

**Figure 8.** Relative importance degree of indices applied in the classification process.

### 4.4. Assessment of RF Performance

In this study, the performance of the RF model was evaluated using various statistical measures, such as RMSE, MAE, OA, KC, and ROC-AUC.

As seen in Figure 9, the variation of false-positive rates versus true positive rates was illustrated using the ROC-AUC. According to Figure 9, the ROC-AUC value for the model generated by the RF algorithm is equal to 0.91, which indicates high accuracy in the generated model. Furthermore, the KC and OA obtained 0.87 and 91.11%, respectively, demonstrating the satisfying quality of RF performance. Therefore, based on the results, it seems that the application of the GEE (implemented in the CCP) and RF model is highly efficient when preparing flood risk mapping. In addition, to further ensure the accuracy of the model, RMSE and MAE values returned by the training stage obtained 0.195 and 0.26, respectively, while for the testing stage, these values were equal to 0.25 and 0.31. Therefore, by performing the above analyses and by measuring the performance of the computational models, it can be concluded that the application of RS along with the GEE platform is a very useful tool when determining effective indices for the occurrence of floods. Moreover, the findings of this research can be important to relevant for the straight flood control initiatives of region authorities at the basin-scale to minimize vulnerability to flooding, improve early flood alarm systems, and evacuate flood victims.

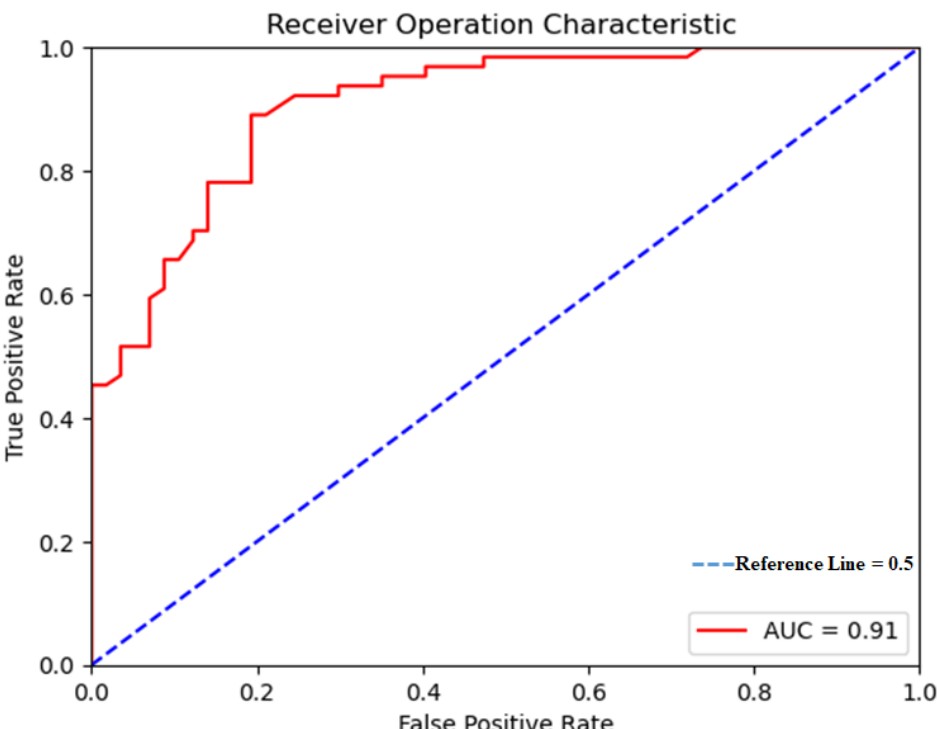

**Figure 9.** Results of AUC-ROC method in validating RF.

*4.5. Comparison of Results with Previous Studies*

The comparison of present results with the previous literature was made efficiently in terms of the accuracy level of predictive tools, the usability of the risk indices, and the typical selection of satellites.

Feng et al. [46] employed Unmanned Aerial Vehicle (UAV) RS for flood monitoring. In their study, the RF model (OA = 87.3%) had better performance than maximum likelihood and ANN. The present study showed that the usability of Landsat 8 satellite imagery was more efficient for flood monitoring than the UAV. Landsat 8 satellite imagery is also superior to images taken by the UAV system. Images taken by the UAV with a spatial resolution of less than 5 cm (depending on the flight altitude) make it possible to prepare a flood risk mapping and to set up an emergency response system for receiving very useful information during a flood. However, images taken by Landsat 8 are superior to UAV images in flood risk mapping. Creating UAV images is very expensive, whereas Landsat 8 images are provided to users for free. As a major advantage, Landsat 8 satellite imagery benefits from historical data at different times, whereas UAV images do not exist in this regard. In the same area, the amount of data processed in the UAV is significantly higher than that of the Landsat 8 satellite imagery. Landsat 8 images have different spectral bands to monitor flood and its risk levels, making it possible to produce spectral indices. Although there are UAVs with multispectral sensors, the data processing volume is gently on the rise, particularly when using a large-scale.

In the Youssef et al. [58] study, the AHP method was utilized for flood risk mapping. According to the evaluation results, the OA obtained 83 percent compared to the OA = 91.11 of the present study. Moreover, the presented method is relatively accurate compared to other ML methods. Additionally, Eini et al. [38] examined flood risk mapping using the MaxEnt and GARP methods. Based on the results, the KC for MaxEnt and GARP was equal to 0.82 and 0.86, respectively, which obtained excellent accuracy compared to the results of the present research with KC = 87. In addition, Soltani et al. [23] prepared an FRM utilizing the NDVI index and rainfall. Based on the evaluation of the generated map, the average value of MAE achieved = 5.076, indicating low accuracy compared to the present results (MAE = 0.31). Compared to the present results, the lack of precision level in the Soltani et al.

(2021) study might likely be due to ignorance towards other environmental indices such as El, RD, Sl, and ST. They applied an improved version of the GMDH model with more polynomial regression equation complexity for flood risk projection compared to the RF model in this study. In the case of flood risk mapping, Soltani et al. [23] used MODIS (or Moderate Resolution Imaging Spectroradiometer) satellite data. The Landsat 8 satellite has a very high spatial resolution compared to the MODIS sensor, which leads to increased accuracy when mapping vegetation and water areas. In addition, Landsat 8 bands are designed to image in the visible, near-infrared, short-infrared, and thermal infrared spectral ranges that can be used to generate effective flood occurrence indices. On the contrary, in order to prepare a flood risk mapping on a global scale, MODIS sensor images are more efficient than Landsat 8 satellite imagery. However, on a local scale, MODIS sensor images are not suitable for flood risk mapping. In fact, L8 satellite images are highly effective in mapping the surface features due to their high spatial resolution.

The statistical performance of the RF model demonstrated a permissible accuracy level (RMSE = 0.25 and MAE = 0.31) when compared to the investigation of Baig et al. [17], which aimed to provide flood risk mapping by means of SVM (SVM–Polynomial Kernel Function (RMSE = 0.466 and MAE = 0.342), SVM–Radial Basis Function (RMSE = 0.455 and RMSE = 0.367)) and GPR (GPR–Polynomial kernel function (RMSE = 0.413 and MAE = 0.373), and GPR–Radial Basis Function (RMSE = 0.450 and RMSE = 0.443)) models. In fact, Baig et al. [17] used hydro-environmental properties (El, SA, DD, TWI, ST, rainfall, distance to stream networks, plan curvature, profile curvature, Land Cover (LC), lithology, Steam Power Index (SPI)) from the Sentinel-1 satellite for flood monitoring. Optical L8 images have a unique advantage over radar images. Radar images from Sentinel-1 are an effective tool for mapping flood areas; however, due to the lack of diverse spectral information compared to L8 satellite imagery, they are not effective for flood risk mapping. Moreover, it is not possible to generate various spectral indices such as NDWI and NDVI using Sentinel-1 radar images. Meanwhile, L8 images are used to produce multiple spectral indices.

As a limitation to this study, lithology, SPI, and LC were not used as risk indices. In addition to the usability of the 11 indices, the study of RFM can comprehensively be performed by considering three indices (i.e., lithology, SPI, and LC). Additionally, Avand et al. [53] applied RF (91% accuracy) and a Bayesian generalized linear model (85% accuracy) to investigate flood risk using Sentinel-1 satellite imagery. As a restriction, Avand et al. [53] could not use the ecological indices of NDVI and NDWI to monitor floods. As such, this issue might be a key cause of computational errors when compared to the present results. In the case of Deep Convolutional Neural Networks (DCNNs) application, the statistical results of Dong et al.'s [68] investigation for monitoring summer floods (using Sentinel-1 satellite imagery) are comparable to the RF model in the present study. Although DCNNs are precise methodologies for flood monitoring, the general structure of RF is simpler and is a less time-consuming model.

Since the performance of the proposed method depends on training examples, it is not possible to implement the method in areas without training examples. On the other hand, in the present study, it is not possible to identify flood-prone areas using Real Time (RT) measurements or flood-prone areas determined by Near Real Time (NRT) measurements. However, in the present study, an attempt was made to use upgraded data to determine flood-prone areas, but this leads to the lack of effective indices in the times before the flood. In addition, there is no possibility to generate spectral indices using optical images in areas where the cloud is the predominant phenomenon (this is not the case for the study area in the present study). One of the major disadvantages of the proposed framework is the complexity of producing effective indices in determining flood risk areas such as the depth index of waterways, canals, and rivers in the GEE platform. However, by increasing the data and making it available to users, this platform can be reliable literature for future analysis.

## 5. Conclusions

Given that extreme climate events such as flooding could improve, in the days ahead, damage to infrastructure may swell, which is likely to increase economic losses. Hence, it is critical to develop a method to assess associated meaningful socio-economic losses. Flood risk mapping is an efficient way to predict and analyze spatial risks; however, such mapping has a complex and systematic process that involves nonlinear and high-dimensional data. Processing this data requires very powerful PC components and a high computational time. To solve this drawback, a web-based platform called GEE was used. To map the flood risk, 11 important risk indices were used. To combine the risk indices, a very common model called RF was used in the open-source interactive Python and GEE interaction package. A number of 400 samples data samples were used to train and test the model, which was recorded from the floods that occurred in recent years. Among the various indices, WRD, RD, Rainfall, and El, indices accounted for about 68.27% of the total flood risk. Among these indices, the WRD index with about 23.8% of the total risk, shows the greatest impact on the flood. Additionally, the second risk index is the RD index, which includes 15.3% of the total indices. Among them, the lowest flood risk percentage was related to the NDWI index, which only included 1.03% of the total risks. However, NDWI, ST, TWI, SA, LU, NDVI, and SL accounted for about 31.73% of the total risk. In general, four indices, El, Rainfall, WRD, RD, and Rainfall, were the most important indices for creating floods in the study area. However, all of the indices together lead to floods and cause irreparable damage to agricultural lands and products as well as to economic resources and benefits. Due to the homogeneity and heterogeneity of the features in the satellite imagery and DEM, each of the risk indices may present different results in various regions. Therefore, the results show the ability of the RF algorithm to identify the most important and even the least effective risk indices. This not only has a significant impact on the data generation time and final map modeling but also increases the quality of decision-making to prevent disasters. According to the flood risk map, about 18% of the total areas were placed in the highest-risk levels, and 21% of the total areas were grouped into higher-risk levels. About 16% of the total areas were categorized in medium-risk levels, whereas 16% of the total areas had lower-risk levels. Ultimately, 29% of the total areas were located in the lowest-risk areas. The results of this study were high quality, so the value of the ROC–AUC for the generated model by the RF algorithm was equal to 0.91. Moreover, the KC (0.87) and OA (90.11) values indicated the high potential of the RF model. Since GEE cloud computing was used to generate the flood risk indices, this led to increased computational speed, the use of a very large number of satellite images, no need to perform the Radiometric and Atmospheric corrections, and free access to all of the data that were used. However, it seems that the use of the GEE platform and RF algorithm to prepare flood risk mapping and risk management is very efficient and effective. To further ensure the accuracy of the model, two benchmark criteria, RMSE (0.25) and MAE (0.31), were considered. Therefore, by performing analyses and examining the performance of the computational models, it can be concluded that the GEE platform is a highly useful tool in flood risk mapping.

In this study, the present framework includes several advantages. The first benefit is the usability of up-to-date indices in the GEE Cloud Computing Platform (GCCP), which can import and process several hundred data in a very short time. Using this platform, researchers will no longer need powerful computer systems to download, preprocess, and post-process satellite images to perform risk analysis. Additionally, in this platform, the possibility of time-series analysis can be conducted conveniently, while without this platform, time series analysis becomes a difficult, time-consuming, and costly thing. It is necessary to effectively use a wide range of flood risk indices because this issue affects the accuracy level of models when defining risk levels. The quality of decision-making during the flood occurrence is inextricably bound up with the number of effective environmental indices in creating floods. The assessment of presented results created a reference for flood risk management, avoidance, and decreasing natural disasters in the study area. In future

research, it is suggested that SAR (Synthetic Aperture Radar) images and multi-sensor optical data be used simultaneously for flood risk detection. In addition, future research will consider multiple climate models combined with ML algorithms to predict future flood risk scenarios in the study area.

**Author Contributions:** H.F.: Data preparation, conceptualization, methodology, writing—original draft preparation, M.N. writing—review and editing, validation, supervision, investigation, formal analysis. All authors have read and agreed to the published version of the manuscript.

**Funding:** This research received no external funding.

**Institutional Review Board Statement:** Not applicable.

**Informed Consent Statement:** Not applicable.

**Data Availability Statement:** Some or all of the data, models, or codes that support the findings of this study are available from the corresponding author upon reasonable request.

**Acknowledgments:** Authors greatly appreciate reviewers for giving constructive comments to improve quality of paper in terms of literature review, methodology analysis, and comparison of the present results with previous investigations in a similar way.

**Conflicts of Interest:** The authors declare no conflict of interest.

## Appendix A

The link to the script that was used to create the flood risk indices directly was provided in the specific format of the Google Earth Engine platform. This script can be found through the following link:

https://code.earthengine.google.com/5daa22a7670e049433ef0ba559eaeeff

In order to run the script, after subscribing to the Google Earth Engine platform, paste the relevant link in the address bar and wait for the script to run. In order to view the results, it is necessary for the user to activate the desired layer tick. This script has been adjusted according to the conditions of the study area in the present study, and it is necessary to change the parameters when considering other areas. After performing the script and viewing the results, the user needs the output of the desired layers in the vector or raster format.

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
