# Peer review of "Flood Risk Mapping by Remote Sensing Data and Random Forest Technique"

_water, doi:10.3390/w13213115_

Round 1

Reviewer 1 Report

Thank you very much for the authors' willingness to revise the manuscript according to the previous comments and suggestions. After reading the revised manuscript, regretfully, the reviewer still has to “Reject” the manuscript again. 

Most of the reviewer’s comments are the same as previous comments:

  1. English needs to be improved. Some sentences are confusing (ex: line 35-37, etc)
  2. The authors failed and need to identify the main novelty of the paper following the state of the art. There are no strong arguments or reasons to develop the proposed model. Lack of comparisons between the existing model and the proposed model. 
    1. From the reviewer’s point of view, the title of this manuscript is misleading. Based on the reviewer’s understanding, the main point of this manuscript is to use the Random Forest algorithm which has never been used for flood risk mapping (lines 134-136).
    2. Line 138: what is the meaning of ‘up-to-date web data’? In the reviewer’s view, all the models that the authors mentioned surely will use up-to-date data. 
    3. Line 138: please define what is ‘strong hardware system’?
    4. Line 139-140: ‘no need for complex and heavy calculations compared to other ML methods’. Emphases this statement. In the reviewer’s view, the main goal for most of all the algorithms is to make the computations or codes more efficient for providing the results. 
    5. Line 140-141: ‘reducing the involvement of human factor in decision-making’ how the proposed model could do this?
    6. Line 150: ‘The main aim of this paper is to prepare a flood risk in Galikesh River basin…’ again in the Reviewer’s view the main goal of this manuscript is not clear, either the authors try to develop a new rapid flood model or to apply a certain model in a certain area. This is very crucial since it will define the methodology of the research. Based on the reviewer’s understanding, the main goal of this manuscript is to develop a new rapid flood model. It can be seen from chapter 4 where the authors focused on comparing the proposed model’s results with other models’ results. The explanation about the flood risk map itself is almost nothing. Therefore, again, in the reviewer’s view, the title is inappropriate. 
  3. Line 57-60, the authors explained that combining RS and GIS is not feasible and time-consuming. From the reviewer’s point of view, some models use this technique and very fast. Perhaps, the authors could provide some researches to support this argument. 
  4. In the reviewer’s view chapter 2.2 is not necessary instead emphasizes more about the case study area. Explain why it is necessary to analyze this area. Add more information into it, e.g. is there any historical flood event occur in the past, what is the area, etc. 
  5. There is no explanation of how the proposed model work. In addition, there is no explanation about the flow chart in Figure 2. In the reviewer’s view, this is very important for the viewer to understand how the proposed model works. 
    1. There is no explanation about the input and output from the proposed model. Is rainfall is one of the inputs? If yes, then there is no information or explanation about how big is the rainfall. In lines 379-380, the authors stated that in this study three flood events were used. 
  6. As indicated in line 253, the results will be evaluated using the collected sample. Please explain what kind of samples that the authors use. Line 378: the sample was divided into two categories, flooded and non-flooded. Please define when it is called flooded or non-flooded.
  1. Repeated sentences were found in lines 248-250 and lines 134-136. In these sentences, the authors claim a baseless statement about RF will, without a doubt, produce a very accurate flood risk map even though in the same sentence the authors also mentioned that RF has never been used for flood risk mapping. 
  2. Line 381-383: use a quantitative measure to define the risk level. It is not clear how to differentiate between very low risk - low risk, and medium risk-high risk. The category that the authors used were only flooded and non-flooded. 
  3. It is not clear how the authors evaluate the model’s performance. In lines 423-424, one of the criteria that authors use is ROC-AUC however there is no further explanation or equation.
  4. The authors claimed one of the main advantages of the proposed model is efficiency. How much time for this model to compute? How much time did authors take to do the pre-post processing computation? How does it compare to the other models, is it faster or slower? What is the computer’s spec that authors used to develop and run this model? Based on this manuscript, the reviewer could not see how this proposed model is more efficient compare to the other models.

Author Response

Most of the reviewer’s comments are the same as previous comments:

C#1:English needs to be improved. Some sentences are confusing (ex: line 35-37, etc)

R#1: Writing expression of the paper was significantly enhanced and grammatical mistakes were removed from the paper.

C#2: The authors failed and need to identify the main novelty of the paper following the state of the art. There are no strong arguments or reasons to develop the proposed model. Lack of comparisons between the existing model and the proposed model. 

R#2: Reviewer’s attention to the introduction section. Literature review, motivation, novelty, and research organization were completed revised and enhanced. Please view this section.  

C#2.1:From the reviewer’s point of view, the title of this manuscript is misleading. Based on the reviewer’s understanding, the main point of this manuscript is to use the Random Forest algorithm which has never been used for flood risk mapping (lines 134-136).

R#2.1: We changed title and added Random Forest classifier to the titles. We revised introduction completely. Please view this section.

C#2.2: Line 138: what is the meaning of ‘up-to-date web data’? In the reviewer’s view, all the models that the authors mentioned surely will use up-to-date data. 

R#2.2: This means “upgraded web-based data” and was added to the paper. Please view revised paper.

C#2.3: Line 138: please define what is ‘strong hardware system’?

R#2.3: This should be changed by “powerful computer hardware/high-end PC components/powerful PC components” and was added to the paper.  Please view revised paper.

C#2.4: Line 139-140: ‘no need for complex and heavy calculations compared to other ML methods’. Emphases this statement. In the reviewer’s view, the main goal for most of all the algorithms is to make the computations or codes more efficient for providing the results. 

R#2.4: This statement was justified. Please view the revised paper.

C#2.5: Line 140-141: ‘reducing the involvement of human factor in decision-making’ how the proposed model could do this?

R#2.5: This statement was removed.

C#2.6: Line 150: ‘The main aim of this paper is to prepare a flood risk in Galikesh River basin…’ again in the Reviewer’s view the main goal of this manuscript is not clear, either the authors try to develop a new rapid flood model or to apply a certain model in a certain area. This is very crucial since it will define the methodology of the research. Based on the reviewer’s understanding, the main goal of this manuscript is to develop a new rapid flood model. It can be seen from chapter 4 where the authors focused on comparing the proposed model’s results with other models’ results. The explanation about the flood risk map itself is almost nothing. Therefore, again, in the reviewer’s view, the title is inappropriate. 

R#2.6: Reviewer’s attention to the title and research organization is appreciable. Title, literature review, motivation, novelty, and research organization were completed revised and enhanced. Please view this section.  

C#3.Line 57-60, the authors explained that combining RS and GIS is not feasible and time-consuming. From the reviewer’s point of view, some models use this technique and very fast. Perhaps, the authors could provide some researches to support this argument. 

R#3. This statement changed and justified. Please view the revised version.

C#4: In the reviewer’s view chapter 2.2 is not necessary instead emphasizes more about the case study area. Explain why it is necessary to analyze this area. Add more information into it, e.g. is there any historical flood event occur in the past, what is the area, etc. 

R#4: We have rewritten “Research Case Study” by adding more details of the case study such as the definition of the case study, the importance of the case study, and previously-occurred floods in the case study. Please view this section.  

C#5: There is no explanation of how the proposed model work. In addition, there is no explanation about the flow chart in Figure 2. In the reviewer’s view, this is very important for the viewer to understand how the proposed model works. 

R#5: Descriptions of Figure 2 were merged into paper. Please view the revised section.

C#5.1: There is no explanation about the input and output from the proposed model. Is rainfall is one of the inputs? If yes, then there is no information or explanation about how big is the rainfall. In lines 379-380, the authors stated that in this study three flood events were used. 

R#5.1: Reviewer’s attention to the explanation of input-out system, rainfall index, and flood events is appreciable. Comprehensive descriptions about input output data were merged into paper. We also added “Rainfall Index” to the previous index and re-calculated all the computations. Moreover, we added another event (there are now four flood events) to the data and re-analyzed the results.

Please view revised version of paper and related Figures 5&8.

C#6: As indicated in line 253, the results will be evaluated using the collected sample. Please explain what kind of samples that the authors use. Line 378: the sample was divided into two categories, flooded and non-flooded. Please define when it is called flooded or non-flooded.

R#6: Some descriptions about the collected samples were added to the revision for more clarifications.

C#6.1: Repeated sentences were found in lines 248-250 and lines 134-136. In these sentences, the authors claim a baseless statement about RF will, without a doubt, produce a very accurate flood risk map even though in the same sentence the authors also mentioned that RF has never been used for flood risk mapping. 

R#6.1: We improved “Definition of RF model” and remove some previous repeated statements. Please view revision.

C#6.2: Line 381-383: use a quantitative measure to define the risk level. It is not clear how to differentiate between very low risk - low risk, and medium risk-high risk. The category that the authors used were only flooded and non-flooded. 

R#6.2: As reviewer requested, we applied the above-mentioned categories and re-analyzed the results. Please view revision and Figure 6.

C#6.3: It is not clear how the authors evaluate the model’s performance. In lines 423-424, one of the criteria that authors use is ROC-AUC however there is no further explanation or equation.

R#6.3: The process of computing ROC-AUC is too long to present in this study. Please view revision.

C#6.4: The authors claimed one of the main advantages of the proposed model is efficiency. How much time for this model to compute? How much time did authors take to do the pre-post processing computation? How does it compare to the other models, is it faster or slower? What is the computer’s space that authors used to develop and run this model? Based on this manuscript, the reviewer could not see how this proposed model is more efficient compare to the other models.

R#6.4: The comparison of present results with the previous literature were efficiently made in terms of accuracy level of predictive tools (data-mining techniques), usability of risk indices, and typical selection of satellites. In this way, since we do not have time performance of the previous literature we could not make comparisons in a way that such models are faster or slower.

Please view comprehensive comparisons in the 4.5 section.

Reviewer 2 Report

This paper presents methods to derive Flood Hazard Risk indices from Landsat-8/OLI images and SRTM/DEM using Google Earth Engine. Altogether 10 indices were used including El, SL, SA, LU, NDWI, NDVI, TWI, RD, WRD, and ST. These indices were merged together by Random Forest model, which is a decision-tree classification approach. The paper gives some detail on data, methods and some results. However, there are some concerns on the paper:

1. The manuscript does not show any figures. It is hard to see the results.

2. The authors claimed to use ten indices for flood risk mapping, and four indices: WRD, RD, El, and Sl accoutered for about 74 percent of the total flood risk. In fact, for flood risk mapping, the most important indices should be precipitation for floods caused by rainfall, and water equivalent of snow/ice and air temperature for snowmelt floods. Without these variables considered, flood risk mapping is without much value.

3. The authors use random forest classification approach to build the flood risk model. To use decision-tree technique, training samples should be collected. From the manuscript, three flood events were used to define risks with no flooded regions defined as low risk, and flooded areas defined as high risk. It is not clear whether the risk index is in 0/1 binary values, or with continuous distribution from 0 to 100%. For flood risk mapping, three flood events are not adequate for building models.

4. Some indices such as NDWI and NDVI are derived from Landsat-8/OLI imagery. Does it mean when the model is applied for flood risk mapping, Landsat-8/OLI images are required. Considering the long revisiting intervals of Landsat-8 data and cloud issue, how can this model be applied in real-time flood risking mapping?

5. The validation part is very weak. Did the authors apply the model in other flood events? How is the result? Did the model accurately predict the flooded regions?

Overall, the paper is written okay, but the science part is very weak. I would recommend major revision before publishing.

Author Response

Response to the Reviewer#2 Comments:

C#1. The manuscript does not show any figures. It is hard to see the results.

R#1. All the Figures had already been submitted to the journals.

C#2. The authors claimed to use ten indices for flood risk mapping, and four indices: WRD, RD, El, and Sl accoutered for about 74 percent of the total flood risk. In fact, for flood risk mapping, the most important indices should be precipitation for floods caused by rainfall, and water equivalent of snow/ice and air temperature for snowmelt floods. Without these variables considered, flood risk mapping is without much value.

R#2. The Rainfall [Maximum one-day precipitation (M1DP)] index was added to the edited manuscript and evaluated. Accordingly, a new section was added to the paper and Fig.8 “Relative importance degree of indices applied in the classification process” was modified.

3.2.11. Rainfall (M1DP)

     In the Google Earth Engine cloud computing platform, using different satellites, it is possible to extract the trend of precipitation changes Available in 30-minute, 3-hour, daily, and monthly intervals. Climate Hazards Group InfraRed Precipitation with Station data (CHIRPS) is a 30+ year quasi-global rainfall dataset. CHIRPS incorporates 0.05-degree spatial resolution satellite imagery with in-situ measurement data to create gridded rainfall time series for trend analysis and seasonal drought monitoring. In the current research, by analyzing Maximum one-day precipitation (M1DP) index, the former is selected as it is more representative of the precipitation that creates floods. Then the M1DP values of the stations are interpolated by Kriging interpolation tool and the network layer of the M1DP index is finally generated (100100 m) based on the GIS approach.

C#3. The authors use random forest classification approach to build the flood risk model. To use decision-tree technique, training samples should be collected. From the manuscript, three flood events were used to define risks with no flooded regions defined as low risk, and flooded areas defined as high risk. It is not clear whether the risk index is in 0/1 binary values, or with continuous distribution from 0 to 100%. For flood risk mapping, three flood events are not adequate for building models.  

R#3: Reviewer’s attention to this point is appreciable. To clarify this point, we used additional flood event. We are now applying four events. We also widened range of the risk indices to five classes: lowest (0.0–0.2), lower (0.2–0.4), medium (0.4-0.6), higher (0.6-0.8), and highest (0.8-1.0). Hence, all the changes in the revised paper were highlighted and Figure 6 was modified. Please view highlights and modifications in the revised version.

C#4. Some indices such as NDWI and NDVI are derived from Landsat-8/OLI imagery. Does it mean when the model is applied for flood risk mapping, Landsat-8/OLI images are required. Considering the long revisiting intervals of Landsat-8 data and cloud issue, how can this model be applied in real-time flood risking mapping?

R#4. Reviewer’s attention to the NDWI and NDVI are appreciable. Landsat 8 satellite imagery, despite its 16-day temporal resolution, is often difficult to map flooded areas due to clouds and unfavorable weather conditions. If our goal is to map flood-prone areas or flood risk maps, there is no need to use Real-Time images, and we can use Near Real-Time satellite images. Therefore, the purpose of our paper is to map flood-prone areas, not Inundate areas. As you know, in the presence of clouds and unfavorable weather conditions, radar images are used to map flooded areas, and in some cases a combination of radar satellite images such as Sentinel-1 and optical images such as Landsat-8, Sentinel-2, and MODIS are used. May be. Accordingly, the NDVI and NDWI indices can be easily calculated as a time series in the rainy season.

C#5. The validation part is very weak. Did the authors apply the model in other flood events? How is the result? Did the model accurately predict the flooded regions?

R#5. As discussed in detail in the introduction part, the Random Forest algorithm has been used in many risk assessment studies. Some references are provided below. Therefore, due to the increasing volume of manuscripts as well as the lack of access to training and testing data, it is not possible to provide methods in different study areas at this stage.

Breiman, L. (2001). "Random forests." Machine Learning 45(1): 5-32.

Tesfamariam, S., Liu, Z., 2010. Earthquake induced damage classification for reinforced concrete buildings. Struct. Saf. 32 (2), 154–164.

Dong, L.J., Li, X.B., Peng, K., 2013. Prediction of rockburst classification using Random Forest. Trans. Nonferrous Met. Soc. China 23 (2), 472–477.

Deng, W. and J. Zhou (2013). Approach for feature weighted support vector machine and its application in flood disaster evaluation." Disaster Advances 6(4): 51-58.

Markus, I., Clement,A., Tatjana, K. (2012). Tree species classification with random forest using very high spatial resolution 8-band worldview-2 satellite data. Remote Sens. 4, 2661–2693.

Mihailescu, D.M., Gui, V., Toma, C. I., Popescu, A., Sporea, I. 2013. Computer aided diagnosis method for steatosis rating in ultrasound images using random forests. Med. Ultrason. 15 (3), 184–190.

Reviewer 3 Report

Review of the paper:

"Rapid Flood Hazard Risk Mapping by Remote Sensing Data and GIS Techniques in the Google Earth Engine Web-Based Platform: A Case Study of Galikesh River Basin in Northern Iran"

Overview

This paper focuses on flood hazard mapping using a methodology that combines web-based platform (GEE), remote sensing and intelligent learning machine. The authors applied this approach to a case study in Iran. 

General comments

The topic faced by this work is interesting and meets the aims and scope of the journal. I think that this work has the potential to become an interesting work, due to the combination of techniques that represent the core of the methodology. I've appreciated also the analysis of web-based platform (GEE) in this field. However, according to my standard, the quality of the presentation is very low. Several parts are quite confused and it is quite hard to understand what the authors want to stress. Moreover, I cannot understand the novelty of the study or the need of this research. The approach proposed by the authors is not discussed against hydrodynamic-based approaches that represent the most reliable method for flood mapping and risk analysis. Therefore, the paper in its present form looks like an academic exercise more than a scientific work in which one can find a real advance for the knowledge in this field. 

Specific comments

1) Introduction is too long and it seems to me very confused. As an example, lines 56-61 I cannot understand what is the point here. These sentences are too vague and no references supported these ideas. Mostly, you do not analyse what should be the core of your introduction. Why I should not  use the common hydrodynamic-based approach for flood mapping and risk analysis is not explained. You don't provide any reference on this, so that your presentation has an evident limitation. In order to help you on this task, I selected 10 recent papers (reported at the end of my review) in which flood hazard mapping is faced from different point of views.  I suggest to use these and similar papers to highlight the limitations of those approaches that can be solved using your methods. Morevoer, you can use them to add a "Discussion" section in which you can provide further consideration on the added values provided by your methodology in the light of the obtained results. This will give more scientific soundness to your work.

2) Section 3: 

- Lines 213-215: It is impossible to understand what do you mean here. This sentence is not supported by anything. Please consider the hydrodynamic-based approch. What are the "combination of various factors with linear and nonlinear variables and the relationship among variable"? 

3) Random forest model is well-known in the literature. Therefore it is not a novelty of this study. Why did you explain it? I consider to reduce this part because its presentation is useless. The readers who don't already know this approach cannot understand anything from this presentation. 

4) Section 3.2. I cannot see anything related to flow velocity that is one the most important factor for flood risk analysis. This information can be extracted by hydrodynamic-based approach. So I'm quite skeptical about the use of your approach for practical risk analysis.

5) Section 4.3. It is very hard to understand what you want to discuss here

6) A section in which you underline the limitation of this work and the impact on the related literature is missing. Please add it

7) Conclusions: the conclusions are limited to the analyzed case study. No general conclusions have been highlighted by the authors. So what is the real advance of this work?

8) After reading your paper, I wonder if the topic of your work is more oriented to flood susceptibility rather than flood hazard mapping. Please consider if this is more proper for your study, starting from the title.

Recommendation

The paper should be rejected due to poor and unclear presentation, lack of focus and motivation. However I fell that a deep revision of the manuscript and the rewriting of several sections of this work, together with a better discussion of the need of this approach in relation to more consolidated methods for flood mapping, might improve the interest of the work. For these reasons, my recommendation is "major revision" because I'm confident that the authors will be able to provide a significant clarification of the added value of their research. 

Cited works

[1] DOI: 10.1016/j.jhydrol.2017.11.036

[2] DOI: 10.1016/j.jhydrol.2016.01.020

[3] DOI: 10.1007/s11069-016-2382-1

[4] DOI: 10.1016/j.advwatres.2016.05.002

[5] DOI: 10.1002/2015WR016954

[6] DOI: 10.1016/j.jhydrol.2021.126306

[7] DOI: 10.1016/j.jhydrol.2019.02.008

[8] DOI: 10.2166/nh.2018.040

[9] DOI: 10.1016/j.envsoft.2020.104889

[10]DOI: 10.1016/j.jhydrol.2019.124406

Author Response

Response to the Reviewer#3 Comments

General comments

The topic faced by this work is interesting and meets the aims and scope of the journal. I think that this work has the potential to become an interesting work, due to the combination of techniques that represent the core of the methodology. I've appreciated also the analysis of web-based platform (GEE) in this field. However, according to my standard, the quality of the presentation is very low. Several parts are quite confused and it is quite hard to understand what the authors want to stress. Moreover, I cannot understand the novelty of the study or the need of this research. The approach proposed by the authors is not discussed against hydrodynamic-based approaches that represent the most reliable method for flood mapping and risk analysis. Therefore, the paper in its present form looks like an academic exercise more than a scientific work in which one can find a real advance for the knowledge in this field. 

Response# Reviewer’s attention to the structure of the introduction section is appreciable. Literature review, motivation, novelty, and research organization were significantly improved. Some descriptions of the introduction were also removed. Please view the newly-structured introduction and highlights.

Specific comments

1) Introduction is too long and it seems to me very confused. As an example, lines 56-61 I cannot understand what the point here is. These sentences are too vague and no references supported these ideas. Mostly, you do not analyze what should be the core of your introduction. Why I should not use the common hydrodynamic-based approach for flood mapping and risk analysis is not explained. You don't provide any reference on this, so that your presentation has an evident limitation. In order to help you on this task, I selected 10 recent papers (reported at the end of my review) in which flood hazard mapping is faced from different point of views.  I suggest to use these and similar papers to highlight the limitations of those approaches that can be solved using your methods. Moreover, you can use them to add a "Discussion" section in which you can provide further consideration on the added values provided by your methodology in the light of the obtained results. This will give more scientific soundness to your work.

Response#1: Reviewer’s attention to the limitation of introduction section is highly appreciable.

Descriptions of numerical models (finite element methods and finite difference methods) and effective variables on the flood simulation were discussed. We used five references. Please view introduction section.

Also, the following paragraph (as seen in the previous version) was removed from the introduction section:

“Clearly, there is still ferocious demand for training big data, the complexity of mathematical relationships, and the very high computational time. In addition, the use of accurate, effective, and up-to-date information plays a substantial role in defining the quality of decision making and the final produced map. However, in the mentioned studies, due to the very high volume of data generation and the need for a strong hardware processing system, data from the past times have been used for flood risk analysis. On the other hand, ML techniques and MCDM methods are comparatively permissible for flood risk mapping, although they have many complexities and difficulties in practical applications. For instance, in MCDM methods such as AHP, the weights of the criteria are selected before making a decision with the intervention of a human factor and using the opinions of experts and specialists active in that field. Therefore, the complexity of these methods, in which the coefficient of the importance of each criterion depends on the decision-making parameters, appears as a technical problem in the flood risk mapping, leading to a decrease in the quality of decision-making in the flood risk map.”

2) Section 3: 

- Lines 213-215: It is impossible to understand what do you mean here. This sentence is not supported by anything. Please consider the hydrodynamic-based approach. What are the "combination of various factors with linear and nonlinear variables and the relationship among variable"? 

Response#2: Reviewer’s attention to this important point is highly appreciable. These sentences were removed.

3) Random forest model is well-known in the literature. Therefore it is not a novelty of this study. Why did you explain it? I consider to reduce this part because its presentation is useless. The readers who don't already know this approach cannot understand anything from this presentation. 

Response#3: During the previous submission, reviewers had recommended that authors need to present full descriptions of the random forest classifier. Anyway, we have diminished the volume of descriptions according to the new round of revision. Please find the revision. 

4) Section 3.2. I cannot see anything related to flow velocity that is one the most important factor for flood risk analysis. This information can be extracted by hydrodynamic-based approach. So I'm quite skeptical about the use of your approach for practical risk analysis.

Response#4:  Reviewer’s attention to the taking the flow velocity into consideration during flood monitoring is appreciable. When we are using remote sensing techniques, we will no longer need to use flow velocity because it is a hydrodynamic/hydraulic variable. To respect the reviewer’s comment, we added another environmental index (Rainfall) was added to the previous indices, then; all the computations were renewed. Please view the revised paper.

 5) Section 4.3. It is very hard to understand what you want to discuss here

Response#5: Reviewer’s attention to the Section 4.3 is highly appreciable. We improved this section. Please view the revised version.

6) A section in which you underline the limitation of this work and the impact on the related literature is missing. Please add it

Response#6: A discussion underlying the limitation of this work and the impact on the related literature were merged into the paper. Please view the revision.

7) Conclusions: the conclusions are limited to the analyzed case study. No general conclusions have been highlighted by the authors. So what is the real advance of this work?

Response#7: Advantages were merged into the conclusion section. Please view the revised version.

8) After reading your paper, I wonder if the topic of your work is more oriented to flood susceptibility rather than flood hazard mapping. Please consider if this is more proper for your study, starting from the title.

Response#8: Reviewer’s attention to the “flood” concept is appreciable. We revised the paper by “flood risk mapping”. Please view whole the paper.

Recommendation

The paper should be rejected due to poor and unclear presentation, lack of focus and motivation. However I fell that a deep revision of the manuscript and the rewriting of several sections of this work, together with a better discussion of the need of this approach in relation to more consolidated methods for flood mapping, might improve the interest of the work. For these reasons, my recommendation is "major revision" because I'm confident that the authors will be able to provide a significant clarification of the added value of their research. 

Response# Thank you for mentioning these notable references. We used these references in introduction. Please view descriptions of these research works through the paper.

Cited works

[1] DOI: 10.1016/j.jhydrol.2017.11.036

[2] DOI: 10.1016/j.jhydrol.2016.01.020

[3] DOI: 10.1007/s11069-016-2382-1

[4] DOI: 10.1016/j.advwatres.2016.05.002

[5] DOI: 10.1002/2015WR016954

[6] DOI: 10.1016/j.jhydrol.2021.126306

[7] DOI: 10.1016/j.jhydrol.2019.02.008

[8] DOI: 10.2166/nh.2018.040

[9] DOI: 10.1016/j.envsoft.2020.104889

[10]DOI: 10.1016/j.jhydrol.2019.124406

Round 2

Reviewer 3 Report

I think that the paper has been significanly improved during the review process. In particular, the authors have clarified several parts of their work and provided suitable modifications according to my suggestions.

Therefore, I believe that the paper has reached a sufficient level of quality and the overall presentation of this research is fairly good. 

I only suggest the authors to check carefully the English languge and style in their final submission, that is the reason why I suggest minor revision.

Author Response

Editors and Reviewers are appreciated giving another opportunity the authors to under take minor revisions. We tracked changes all the corrections made in the revised version for the review. Additionally, we presented figures at the end of paper after tables. In the previous round of revision, we had already submitted figures.

This manuscript is a resubmission of an earlier submission. The following is a list of the peer review reports and author responses from that submission.

Round 1

Reviewer 1 Report

The paper addresses an interesting topic using GEE for rapid flood hazard risk mapping. The authors used GEE to support flood hazard risk mapping in ArcMap and Python environments. Landsat 8 and SRTM data were used to compute ten indicators in the GEE platform. In addition, they proposed a method to compute the importance of the parameters in producing the flood hazard risk map. In general, the paper is written in a good structure and properly explained the results. However, there are important points that must be improved in the manuscript.

  • The methodology is not clearly summarized in the abstract and it might be misleading.
  • The literature review is limited and should be extended by analyzing the works done in GEE and Random forest classifier.
  • It is not clear which method is used to compute the importance of the parameters. It must be revised. Refer to the comments attached to the pdf version of the paper for more details.
  • Conclusion section should be improved by providing conclusions and not only summarizing the study. For example, what are the pros and cons of using GEE in your study.
  • I cannot find the Appendix.

I also provided detailed comments on the pdf version of the manuscript with notes.

Reviewer 2 Report

The paper “Rapid Flood Hazard Risk Mapping with Google Earth Engine Web-based Platform: A Case Study of Galikesh River Basin in Northern Iran” focused on using a web-based platform called Google Earth Engine (GEE) to identify flood hazard risk. Two kinds of satellite images were used and eight different indicators such as Land use, Normalized Difference Water Index, and etc. are considered to produce flood hazard maps. The reviewer thinks the idea of the paper is of interest to the readers of this journal. However, the reviewer recommends for “Reject” of the manuscript under its present format. A lot of content in this paper is actually redundant and details of the study are not clear. From the reviewer’s point of view, the methods involved in this study are not innovative. However, GEE has got a lot of attention recently and the reviewer actually wants to see the details of this study and make a decision if the paper is meaningful to the readers. Some major comments for the authors to be addressed in the revised manuscript before submitting for further consideration.

Major comments:

  1. This paper’s title is “with Google Earth Engine (GEE) Web-based platform”. The reviewer found that many details are not related to GEE. The title may be appropriate if the title is changed to “using satellite images” instead of “with GEE”. However, if the title is changed to “using satellite images”, many studies have focused on the same application to identify flood hazard maps. The authors should explain what is the difference between this manuscript and others. From the current manuscript, the reviewer cannot see the difference.
  2. The authors should condense the content in the introduction section. For example, lines 92 to 120 addressed AHP, AUC, ROC, and others. Lines 121 to 132 are about the applications of data-driven techniques. Should these two paragraphs be merged to one which is more related to the topic and meaningful to readers?
  3. The reviewer did not understand what is the meaning of the sentence in line 131 “However, data from the past times have been used in most studies in the field.” This manuscript also used data from past times to predict flood hazard maps. The reviewer did not know what is the meaning of the sentence.
  4. The authors identified two different sources of satellite images: SRTM DEM and Landsat-8. What are the spatial and temporal resolutions of SRTM DEM? Are they the same as Landsat-8?
  5. The same concerns are given to the indicators: EI, SL, SA, LU, NDWI, NDVI, TWI, RD, WRD, and ST. Different indicators have different temporal and spatial resolutions Are they the same? Is there a normalized process for these data before input to RF? Also, what are the temporal and spatial resolutions for the flood events? All of the details are not found in the current content. For example, if the monthly average value of NDWI is used, is it reasonable to flood events within a day (i.e., three flood events used in the study)?
  6. The authors used three historical flood events to train the RF. What are the standards the authors used to identify “very low”, “low”, “medium”, and “high risk”? The authors mentioned 70% of the data were used for training. Are the remaining 30% of the data used to calculate the performance indexes?
  7. From the reviewer’s point of view, the performance provided by the authors is not convinced. It is because no details are related to how the performance indexes are obtained. Are they compared with observations? If the RF is used to identify “very low”, “low”, “medium”, and “high risk”, how the RMSE and MAE were calculated.
  8. Finally, the reviewer did not think this is a scientific paper unless the details and innovation of the paper are clear to the readers. Therefore, the reviewer suggests for “Reject” of the manuscript under its present format and recommends the authors add details in the content and resubmit to the journal.